# Unveiling pelagic-benthic coupling associated with the biological carbon pump in the Fram Strait (Arctic Ocean)

Simon Ramondenc [1,2] ✉, Damien Eveillard [3,4], Katja Metfies [1], Morten H. Iversen [1,2], Eva-Maria Nöthig [1], Dieter Piepenburg[1], Christiane Hasemann[1] & Thomas Soltwedel [1]

Settling aggregates transport organic matter from the ocean surface to the deep sea and seafloor. Though plankton communities impact carbon export, how specific organisms and their interactions affect export efficiency is unknown. Looking at 15 years of eDNA sequences (18S-V4) from settling and sedimented organic matter in the Fram Strait, here we observe that most phylogenetic groups were transferred from pelagic to benthic ecosystems. *Chaetoceros socialis*, sea-ice diatoms, Radiolaria, and Chaetognatha are critical components of vertical carbon flux to 200 m depth. In contrast, the diatom *C. socialis* alone is essential for the amount of organic carbon reaching the seafloor. Spatiotemporal changes in community composition show decreasing diatom abundance during warm anomalies, which would reduce the efficiency of a diatom-driven biological carbon pump. Interestingly, several parasites are also tightly associated with carbon flux and show a strong vertical connectivity, suggesting a potential role in sedimentation processes involving their hosts, especially through interactions with resting spores, which could have implications for pelagic-benthic coupling and overall ecosystem functioning.

Polar marine ecosystems are exceptionally vulnerable to global warming[1]. In the Arctic, sea-ice extent rapidly declined between 1979 and 2018, particularly during September, with an average decrease in sea-ice cover by 12.8% per decade[2]. This drastic and rapid sea-ice decline had cascading effects on physical (e.g., stratification, light availability), biological (e.g., plankton phenology), and biogeochemical (e.g., carbon cycle) processes[3].

The biological carbon pump is an important carbon sequestration mechanism where particulate organic carbon (POC), produced by photosynthetic fixation in the upper ocean, is exported to the deep ocean and potentially stored in the seafloor for centuries to millennia[4–6]. In addition to carbon export, sinking aggregates represent the main pathway for the downward transport of energy and

taxonomic lineages[7,8], coupling pelagic and benthic diversity. Compared to temperate and tropical regions, the pelagic-benthic coupling is assumed to be particularly tight in the Arctic Ocean due to the strong seasonal pattern of primary production and particle export[9]. While pelagic-benthic interactions are limited during the "pre-bloom" and "bloom" phases, they are very pronounced during the post-bloom period[10]. However, little is understood about how this coupling has changed in recent decades and how it might continue to change.

The dawn of improved and more affordable sequencing technologies has enabled researchers to discover formerly unknown organisms and gain insights into critical processes such as carbon and nitrogen cycles[11], microbial interactions[12], and evolutionary adaptations in marine ecosystems. In parallel with biotechnological advances,

[1]Alfred Wegener Institute, Helmholtz Centre for Polar and Marine Research, Bremerhaven, Germany. [2]MARUM, Center for Marine Environmental Sciences, University of Bremen, Bremen, Germany. [3]Nantes Université, Ecole Centrale Nantes, Nantes, France. [4]Research Federation for the Study of Global Ocean Systems Ecology and Evolution, FR2022/Tara Oceans GOSEE, Paris, France. ✉e-mail: simon.ramondenc@imev-mer.fr

new conceptual and methodological statistical frameworks have been developed to reduce scaling from genes to the ecosystems, thereby enhancing the predictive power of ecosystem models[11,13]. Building on these technical approaches, global data collections provide a further step toward a holistic view of marine ecological biodiversity and connectivity in the upper water column[14,15]. However, these global surveys call for replication at the mesoscale and following longitudinal studies.

At the Long-Term Ecological Research (LTER) observatory HAUSGARTEN, eDNA-based biodiversity studies investigated microbial (prokaryotic and eukaryotic) biodiversity across different habitats, including the processes that govern marine ecosystem functionality and carbon flux[16,17]. LTER HAUSGARTEN is located in the Fram Strait, a hydrographically highly dynamic passage between Northeast Greenland and Svalbard (Fig. 1). In the eastern Fram Strait, the West Spitsbergen Current carries relatively warm and nutrient-rich Atlantic Water northwards to the central Arctic Ocean, while in the western Fram Strait, the East Greenland Current transports cold ice-covered, less saline Polar Water southwards[18]. Hence, this relatively small area provides an excellent opportunity to study how Arctic marine environmental conditions, including bathymetric gradient and sea-ice dynamics, shape the distribution and abundance of polar and temperate species at different temporal and spatial scales. Over the past decade, eDNA metabarcoding analyses at LTER HAUSGARTEN have shown that the two major water currents in the Fram Strait harbor distinct bacterial, archaeal, and eukaryotic microbial communities[19,20], which themselves exhibit a pronounced seasonality[21]. Furthermore, sea-ice dynamics significantly influence pelagic community composition with implications for bloom phenology and carbon export[3].

Environmental and metagenomic analyses identified key planktonic species that drive vertical carbon export in mesopelagic oligotrophic environments[11], highlighting that the role of prokaryotic and viral biota in the biological carbon pump was underestimated. However, limited data and spatiotemporal sampling resolution have not allowed similar analyses for polar ecosystems. In this context, we aim to integrate pelagic/benthic microbiomes with biodiversity associated with sinking particles to understand how the biological carbon pump affects vertical connectivity. Hence, we use DNA sequences from particulate organic matter collected by sediment traps at 200 m water depth in the eastern Fram Strait from 2000 to 2012. Amplicon sequence variants (ASVs) through metabarcoding and network analysis were employed to (i) uncover community structures and filter out the major lineages associated with carbon export in the upper water layers. This long-term time series effectively captured the dynamics of the plankton community on intra- and interannual scales, helping to overcome challenges associated with difficulties in accessing the area by ship. As the impact of sinking particles on the seafloor communities is largely unknown, we also analyzed interannual changes in DNA sequence abundance in the upper centimeter of sediment cores over 15 years (2003–2018). From the environmental and metabarcoding data, it was possible to (ii) isolate groups of sequences closely related to the carbon content. This enables us to (iii) map the pelagic-benthic coupling in the subpolar region and identify the vertical connectivity between lineages of pelagic and benthic ecosystems associated with carbon sequestration based on sequence and topological network analyses.

## Results
### Planktonic co-occurrence network at the central HAUSGARTEN station

The planktonic co-occurrence network built with the sediment trap material and coupled to the Weighted Gene Co-expression Network Analysis (WGCNA) clustering method captured eleven intrinsic subnetworks (Sn) related to environmental and biogeochemical fluxes (Fig. 2). These subnetworks grouped ASVs with similar patterns from sediment trap samples collected over twelve years at the central HAUSGARTEN station HG-IV (Fig. 1b). Five subnetworks showed high ASV abundance (Sn_3, Sn_4, Sn_9, Sn_10, and Sn_11) with 21.6%, 15.0%, 9.3%, 18.6%, and 18.3% of the total pelagic sequences. Conversely, the subnetworks 1, 2, and 5 to 8 represented only 17% of the total pelagic abundance. Subnetworks 3, 10, and 11 were correlated with POC flux (Fig. 2).

Among the subnetworks related to carbon export, only Sn_10 was negatively correlated with POC flux (Figs. 2a and S1). Its relative abundance showed that it was dominated by the group of parasitic nanoflagellates Pirsonia (34%), dinoflagellates (Dino-group-I and II; 26%), and the order Cryomonadida (7%; Fig. 2b). On a temporal aspect, Sn_10 was negatively correlated with the seasonal cycle (Fig. 2a), with the highest contribution occurring during winter and autumn (Fig. 2c). Conversely, Sn_3 and 11 were positively correlated with carbon export (Fig. 2a). Sn_11 was mainly composed of ASVs that were taxonomically assigned to radiolarians and diatoms, with the Chaunacanthida order (46%) and the Bacillariophyta class (34%; Fig. 2b). The seasonal pattern was significantly positively correlated with this cluster (Fig. 2a),

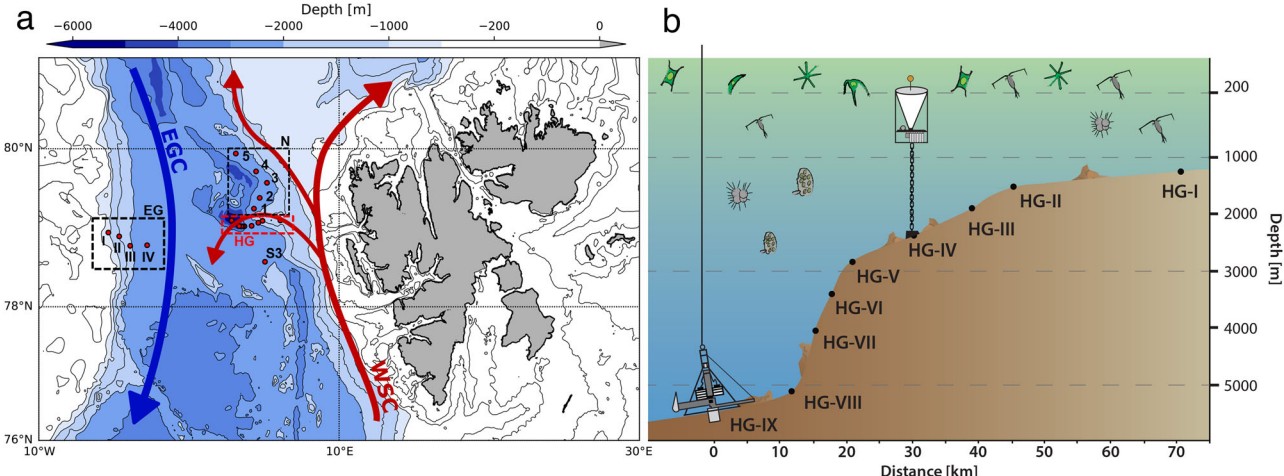

**Fig. 1 | Overview of sampling sites. a** Map of the Long-term Ecological Research (LTER) observatory HAUSGARTEN in the Fram Strait and **b** detail of the longitudinal bathymetric transect from the stations HG-I to HG-IX. The upper water column was sampled at the central station (HG-IV), and the sediment cores were performed at all HG stations. The abbreviations S, N, EG, and HG refer to the South, North, East Greenland, and HAUSGARTEN sampling areas, respectively. The red dots refer to the sampling stations. The red and blue arrows indicate the West Spitsbergen Current (WSC) and the East Greenland Current (EGC) trajectories.

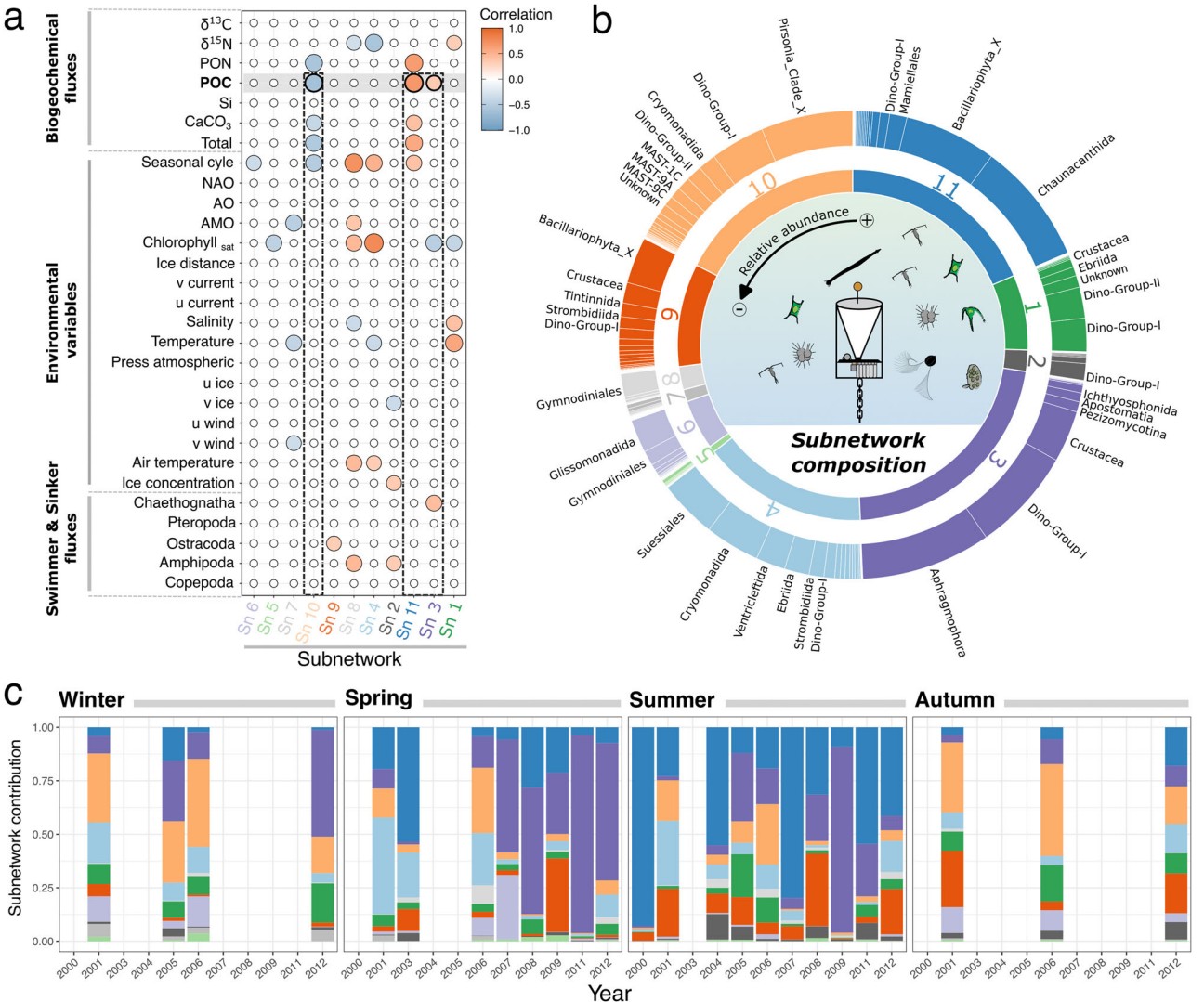

**Fig. 2 | Visualization of the pelagic subnetwork features identified by the WGCNA method. a** Pearson correlations between each subnetwork and the environmental data. **b** Relative abundance of the main orders in each subnetwork. **c** Inter- and intra-annual changes of subnetworks contribution at the central HAUSGARTEN Observatory. Zooplankton and biogeochemical fluxes (including total particulate matter, POC, PON, bPSi, CaCO3, ∂13 C, and ∂15 N) were estimated from sediment traps. Sinkers (i.e., zooplankton passively sink into the trap) and swimmers (i.e., zooplankton actively swam into the trap) larger than 0.5 mm were individually picked, rinsed, identified, and counted under microscope. All environmental variables (such as NAO for North Atlantic Oscillation, AO for Arctic Oscillation, AMO for Atlantic Multidecadal Oscillation, and 'u' and 'v' representing the east–west and north–south components of wind, current, and sea-ice velocity, respectively) included in the correlogram were described in Ramondenc et al.[29]. The colors corresponding to each subnetwork (Sn) are consistent across the subplots.

confirming the highest contribution of Sn_11 during spring/summer (Fig. 2c). From an interannual perspective, Sn_11 showed lowest contributions during spring and summer in 2005 and 2006. Sn_3 was the last cluster that positively correlated with POC flux (Fig. 2a). Aphragmophora representing Chaetognatha species, dinoflagellates, and crustaceans were dominant in Sn_3 with 41%, 32%, and 16% of the sequences, respectively (Fig. 2b). The temporal variability of Sn_3 showed clear seasonal and interannual patterns (Fig. 2c). The highest contribution of Sn_3 was observed after 2006 during spring, which Chaetognatha drove while crustaceans, especially Copepoda, represented summertime of 2009. Interestingly, a strong positive correlation between Sn_3 and chaetognathid swimmers counted manually under the microscope (Fig. 2a) shows a good match between sequencing and microscope counts.

From the two subnetworks that correlated positively with POC, Sn_3, and Sn_11, several ASVs were identified as keystone contributors essential for both carbon export and subnetwork stability, with

stability assessed through the degree centrality of nodes (Fig. 3a). These key ASVs include haptophytes (*Chrysochromulina* sp., *Phaeocystis* sp.), diatoms (*Chaetoceros socialis*, *Melosira arctica*, *Thalassiosira* sp., *Fragilariopsis cylindrus*), dinoflagellates (*Dinophyceae*, *Gymnodinium* sp., *Heterocapsa pygmaea*), chaetognaths (*Parasagitta elegans*, *Eukrohnia hamata*, *Sagitta bipunctata*), and prasinophyte (*Micromonas polaris*).

## Spatial and temporal patterns in seafloor communities at HAUSGARTEN observatory

From the sediment dataset, the co-occurrence network coupled to the WGCNA method captured six intrinsic subnetworks related to abiotic and biotic factors (Fig. 4a) and showing heterogeneous compositions (Fig. 4b). Subnetworks Sn_1, Sn_5, and Sn_6 were the most dominant, representing 44%, 27%, and 14% of the total benthic ASV abundance, respectively. The other three subnetworks represented 15% (Sn_2: 6.9%, Sn_3: 1.1%, and Sn_4: 6.7%).

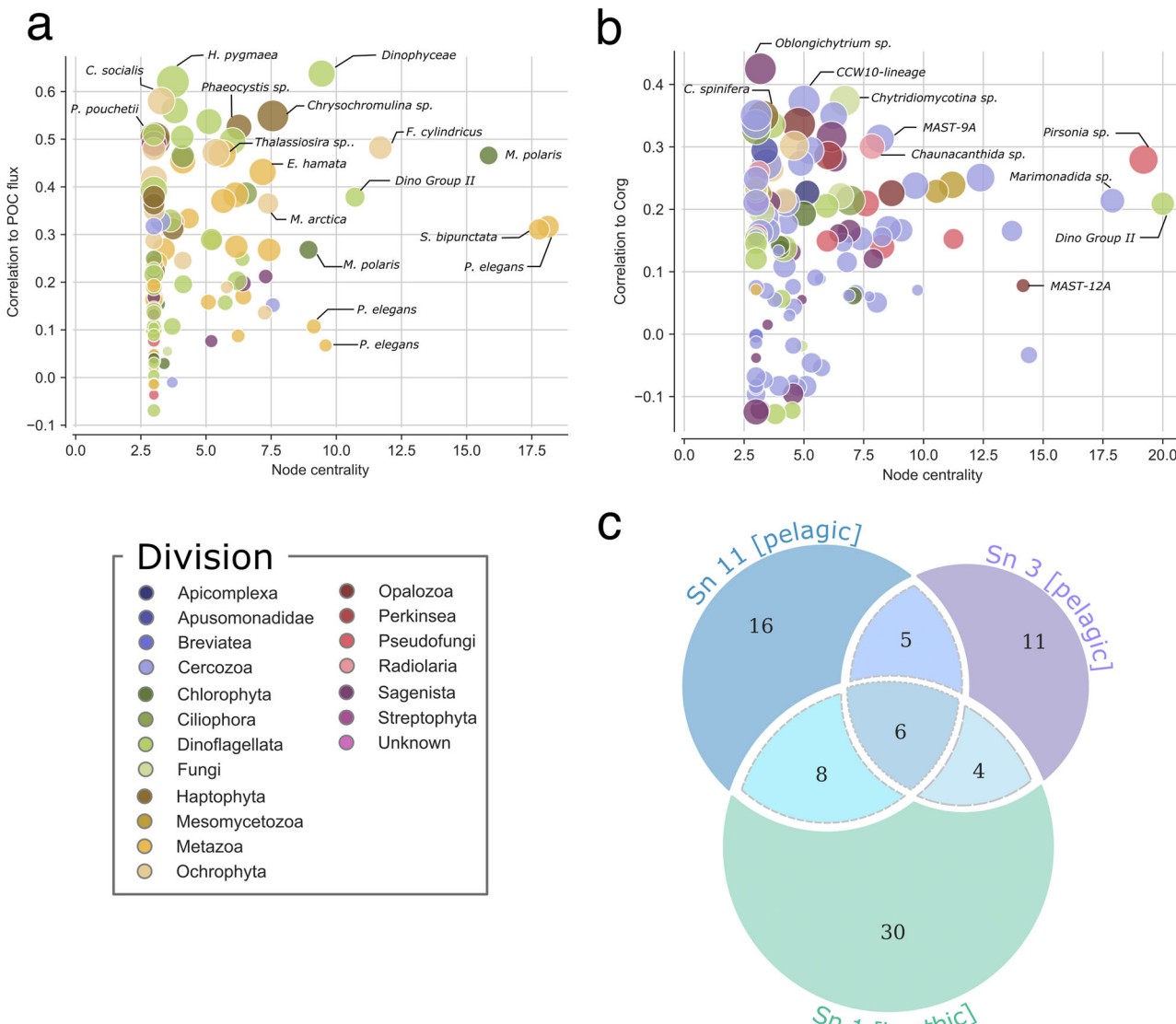

**Fig. 3 | Key taxonomic lineages linked to the carbon cycle at the HAUSGARTEN observatory.** Lineages identification is provided for **a** the pelagic carbon flux and **b** benthic carbon content in the Fram Strait associated with **c** a Venn diagram showing the number of families shared between subnetworks of interest. The degree centrality highlights nodes with the most direct connections, whereas dot color and size represent, respectively, the VIP score in the PLS analyses and taxonomy assigned to the ASVs.

Only Sn_1 was positively correlated with the sediment carbon content and several other factors, such as longitude, diversity indices, bacterial esterase activity, and protein content (Figs. 4a and S2). Diatoms (mainly *Chaetoceros socialis*) largely dominated Sn_1 (46%), followed by Cercozoa (Cryomonadida, Ventricleftida, Marimonadida, and Ebriida), and the heterokont *Oblongichytrium* sp. (Labyrinthulida). Sn_1 dominated at most stations (Fig. 4c), showing the impact of pelagic diatoms on the benthic ASVs. Independent of sample availability, an interannual trend was observed at all HAUSGARTEN sites over the past fifteen years. A cyclic pattern of Sn_1 contribution emerged with three higher phases (2003–2005, 2008–2011, and 2015–2018) interspersed with two periods of decrease (2005–2008 and 2011–2015) (Fig. 4c). Furthermore, the benthic eDNA sequences revealed spatial variability of the subnetworks' contribution to sediment samples in the Fram Strait. Indeed, Sn_1 correlated positively with the longitude, i.e., the sediment cores collected in the eastern Fram Strait, close to Svalbard, contained more carbon and pelagic diatoms than stations in the western Fram Strait.

A weak contribution of Sn_1 to the samples analyzed implied a stronger influence of Sn_6, and vice versa. Sn_6 was mainly composed of Cercozoa (Novel_clade_2, protaspa_lineage, and *Filoreta* sp.), Nematoda (*Halomonhystera* sp.), and Annelida (*Aurospio foodbancsia*) (Fig. 4b). Irrespective of their abundance, several Sn_1 lineages contributed strongly to sediment carbon content (Fig. 3b). Among them, several Cercozoa (e.g., CCW10-lineage), Haptophyta (e.g., *Chrysocampaluna spinifera*), photosynthetic taxa (e.g., dinoflagellates and diatoms), and parasitic lineages (e.g., order Chytridiomycotina, *Oblongichytrium* sp., *Pirsonia* sp.) were correlated with sediment carbon content and/or were decisive for subnetwork stability. The Venn diagram (Fig. 3c) shows that the benthic subnetwork Sn_1 shared more families with the pelagic Sn_11 than the pelagic Sn_3, suggesting a stronger vertical coupling between pelagic Sn_11 and benthic Sn_1.

Despite no significant correlation with sediment carbon content, the last four benthic subnetworks (Sn_2-5) exhibited spatiotemporal patterns. Sn_5 dominated at station HG-V in 2011, 2015, and 2016 and at station N4 in 2009 (Fig. 4c). Various diatoms and flagellates taxa often

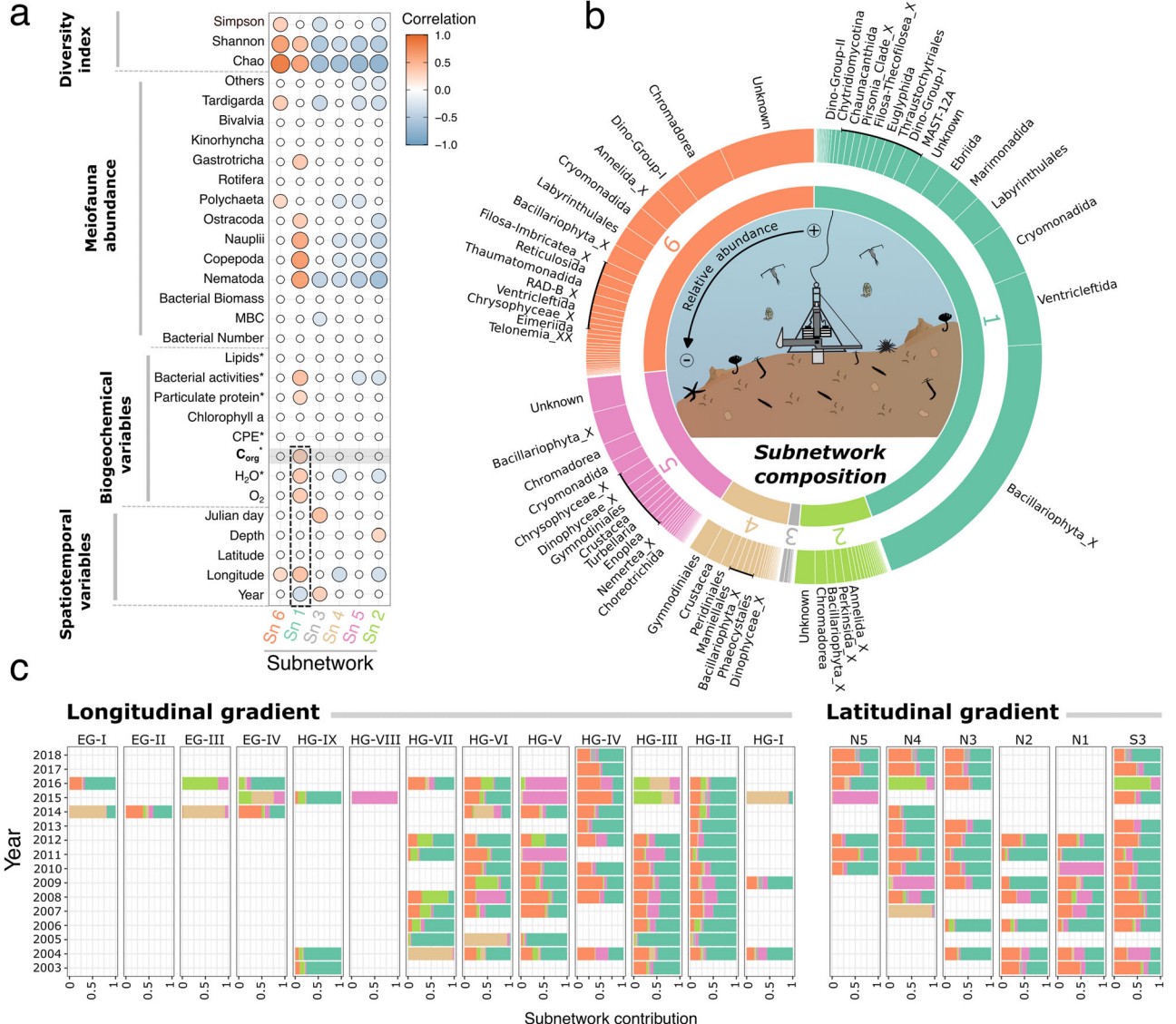

**Fig. 4 | Visualization of the benthic subnetwork features identified by the WGCNA method. a** Pearson correlation between each subnetwork and the environmental data. **b** Relative abundance of the main orders in each subnetwork. **c** Spatiotemporal changes of subnetworks contribution at the HAUSGARTEN Observatory. Acronyms for biogeochemical measurements performed on sediment cores are as follows: MBC (Mean bacterial Biomass per Cell), CPE (chloroplastic pigment equivalents), Lipids (phospholipid concentrations), Corg (organic carbon content), and H2O (water content). An asterisk (*) denotes core parameters averaged (Corg and H2O) or summed (CPE, Lipids, Bacterial activities, particulate protein) over the top five centimeters of the sediment. All meiofauna abundance and biogeochemical variables included in the correlogram are described in the supplementary file. The colors corresponding to each subnetwork (Sn) are consistent across the subplots.

detected in sea-ice and/or sea-ice-associated (*Fragilariopsis cylindricus, Nitzschia* spp., *Navicula* spp., Melosira arctica, Synedra hyperborea, Paraphysomonas foraminifera) dominated in Sn_5. Moreover, ten ASVs associated with nematodes of the class Chromadorea contributed to Sn_5 (e.g., *Leptolaimus* sp., *Desmoscolex* sp.). Contrary to benthic Sn_1 and Sn_6, Sn_2 and Sn_4 were negatively correlated with longitude and eukaryotic ASV richness (Fig. 4a). Sn_2 was composed of sea-ice diatoms (e.g., *Porosira glacialis*), nematodes (e.g., *Sphaerolaimus hirsutus*), and annelids (e.g., *Amphicorina* sp.). In contrast, Sn_4 was dominated by dinoflagellates (e.g., Peridiniales, *Gyrodinium fusiforme*), copepods (e.g., *Calanus helgolandicus*), and haptophytes (e.g., *Chrysochromulina* sp., Phaeocystis pouchetii). Based on the composition and spatial distribution of Sn_2, Sn_4, and Sn_5, we suggest that these subnetworks illustrate the influence of ice cover on benthic ecosystems. Sn_3 was the smallest subnetwork and was more influenced by the sampling period, i.e., the timing of the ship expeditions (Fig. 4a),

and was dominated by ciliates (Strombidinopsidae) and pelagic diatoms (e.g., *Chaetoceros* sp.).

## Pathways of pelagic-benthic coupling based on the sequences and subnetwork topology

The graph-alignment analysis performed in this study was used as a computational approach to represent the pelagic-benthic coupling (Fig. 5). This method allowed us to identify ASVs from two ecosystems that play the same ecological roles within the community structure. For this purpose, we considered two different features and identified the best trade-off between them. Thus, ASVs from distinct ecosystems can be associated based on (i) their taxonomy (i.e., similar or closely related) and (ii) their role within the ecosystem structure (i.e., similar or closely related node centrality in co-occurrence networks). L-GRAAL analysis computes both features and estimates their best consensus via an alpha score (here, an alpha value of 0.4). The lowest centrality

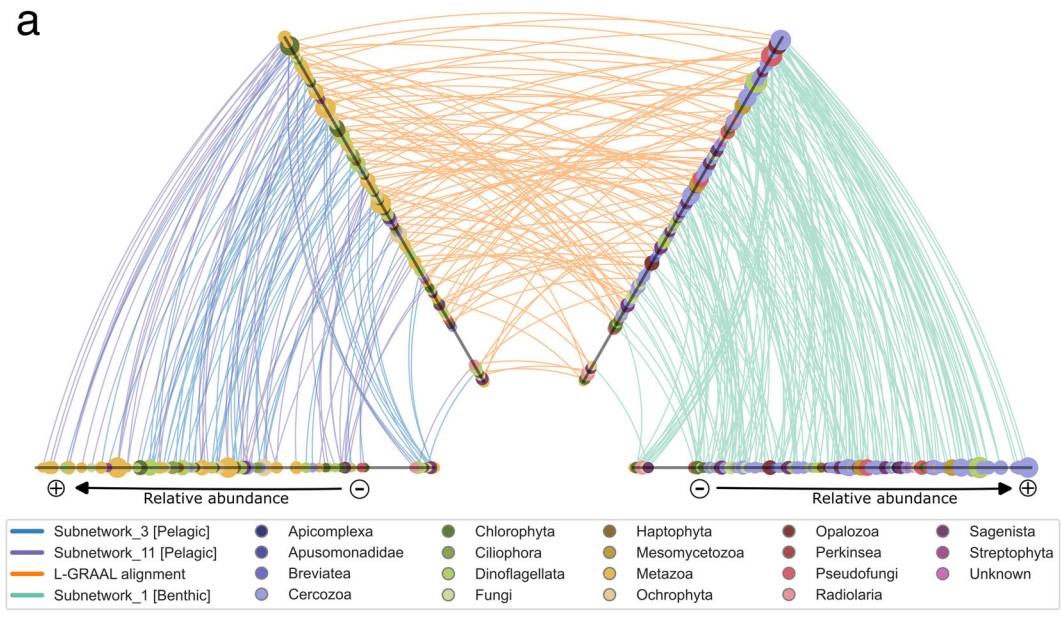

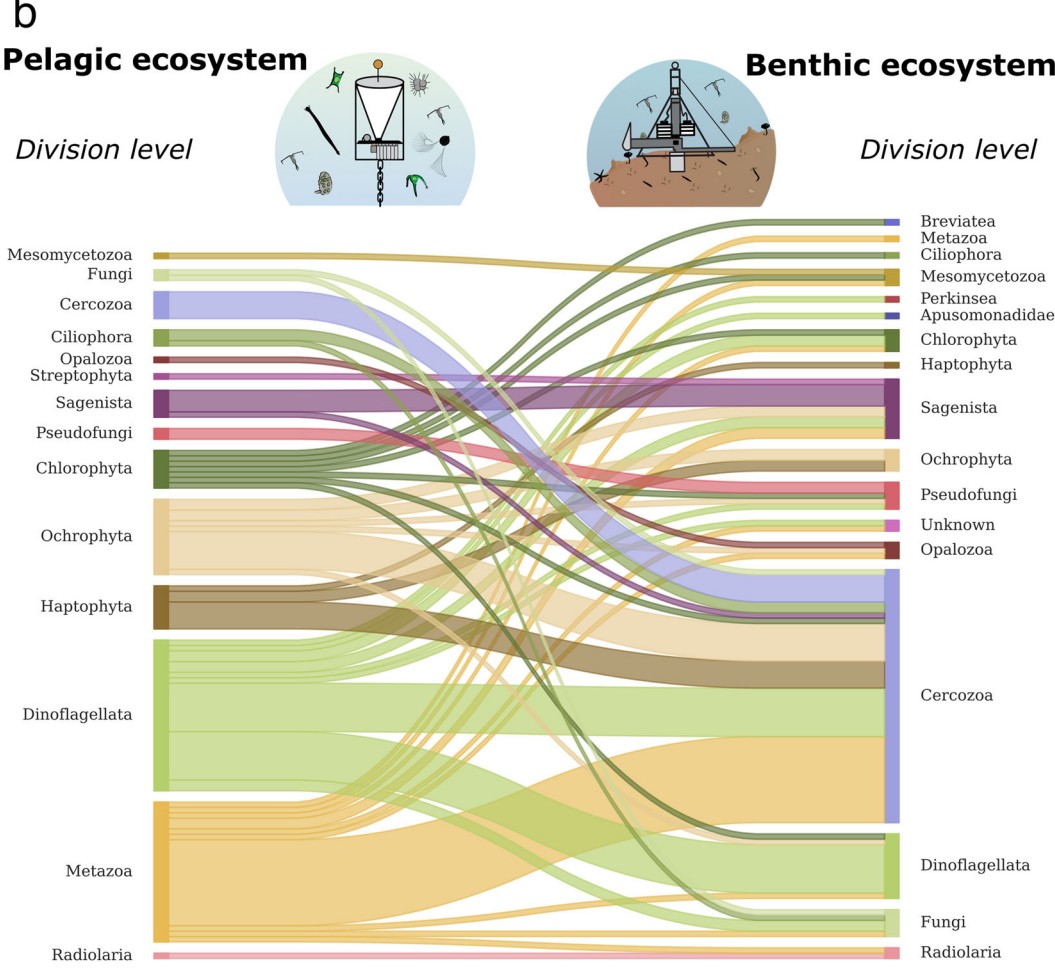

**Fig. 5 | Pelagic-benthic coupling and its partitioning according to the graph-alignment analysis of the subnetworks associated with the carbon cycle.**
**a** Hiveplot representing both the subnetworks co-occurrence (pelagic on the two left axis: Sn_3-purple and Sn_11-blue; benthic on the two right axis: Sn_1-green) and the sequences alignment (orange edges) at alpha 0.4. The nodes on each axis are sorted by their relative abundance, their colors show the taxonomy, and their size represents the centrality of each ASV within the subnetwork 3–11 and subnetwork 1, respectively, for pelagic and benthic ecosystems. Each axis was split into two parts; the internal nodes representing the ASV sequences found in the pelagic and benthic subnetworks and the external nodes for ASV found only in the pelagic subnetworks or only in the benthic subnetwork. **b** Alluvial plot showing the partitioning between pelagic and benthic environments based on the alignment and the taxonomy. Line width indicates the number of ASVs aligned between the two environments, signifying connectivity between those specific phyla.

scores show that the pelagic subnetworks were less self-connected than the benthic subnetwork (Fig. 5a). ASVs with moderate abundance had the highest centrality scores in the pelagic subnetworks. In contrast, benthic subnetwork showed no distinct trends. This indicates high and low benthic-abundant keystone ASVs in the carbon subnetwork. 103 nodes out of 222 possible were aligned between pelagic and benthic subnetworks, and 29% were connected to the same taxonomic division. Only a tiny fraction of ASVs were aligned with themselves: Chaunacanthida, Pycnococcaceae, and the diatom *Chaetoceros cinctus*, suggesting a direct export from the ocean surface to the seafloor via sedimentation. All phyla recorded in both pelagic and benthic zones, except Streptophyta, were connected in the Alluvial diagram (Fig. 5b), providing a simple holistic view of the node connections according to the taxonomy. The most significant differences were observed for Cercozoa and Metazoa, represented by many sequences in the benthic and pelagic ecosystems.

## Discussion

### Temporal changes of pelagic communities and carbon export

Winter and autumn are characterized by light limitation and weak water column stratification. During these seasons, we mainly observed the dominance of parasites such as the nanoflagellate genus *Pirsonia spp.* and dinoflagellates (especially Syndiniales), confirming the findings of previous seasonal studies in the region[21]. While the *Pirsonia* species infect planktonic diatoms[22], Syndiniales are widely distributed in marine environments[14] and can infect many hosts (e.g., dinoflagellates, radiolarians, copepods). Syndiniales group-I, recently observed in high abundance in polar oceans[23], was also the most dominant group in our samples. Therefore, our findings of the negative correlation of Syndiniales and *Pirsonia* with the POC flux at 200 m water depth suggest that winter and autumn were pronounced by high recycling and low carbon export throughout the entire time series. These results align with a recent publication indicating that Syndiniales play a significant role in flux attenuation through remineralization[24].

In spring and summer, irradiance reaches a critical threshold, and stratification is enough to sustain a high diversity of eukaryotic lineages. The high eDNA abundance of diatom lineages observed in spring and summer reflects the annual growth of phytoplankton in the Arctic marine region[25]. Our study revealed that diatoms are necessary for carbon export and that *Chaetoceros* sp. and *Thalassiosira* sp. dominated the material collected by sediment traps. In addition, ice-associated diatoms such as *Melosira arctica and Fragilariopsis cylindrus* were identified as key species for carbon flux to 200 m depth. This result was not unexpected as diatoms contribute up to 40% of the total marine primary production[26] and have been reported as essential contributors to POC fluxes on a global scale[11,27]. Under climate change, rising water temperatures and sea-ice retreat will significantly affect diatom communities[13], with implications for global biogeochemical cycles. In the Fram Strait, a shift from diatoms to *Phaeocystis* as dominant primary producers caused by the warm water anomaly in 2004-2006 has already been observed[28], explaining and confirming the weak contribution of the pelagic Sn_1 and POC flux during this period[29]. Despite their slow sinking velocity[30,31], we observed the haptophyte *Phaeocystis* sp. and the prasinophyte *Micromonas polaris* in both sediment trap material and benthic samples. The occurrence of these lineages in the mesopelagic zone is still an enigma. Several reasons could explain their presence in the deep ocean, such as the downward physical transport of water masses[32,33], repacking of single suspended cells into fast-settling zooplankton fecal pellets[34], or ballasting by cryogenic minerals[35].

Regarding heterotrophic eukaryotes, the eDNA data showed similar phenology and interannual changes of Chaetognatha as described by Ramondenc et al.[29], with higher abundances observed during the spring periods from 2007 to 2012. According to eDNA results, *Parasagitta elegans* was the dominant species. Chaetognatha

lineages in Sn_11 were significantly correlated with POC flux, as previously suggested by Ramondenc et al.[29]. We showed that *Sagitta bipunctata*, *P. elegans*, and *Eukrohnia hamata* were identified as key species for carbon export (i.e., strong correlation and high centrality). This could be because (i) chaetognaths are not strictly carnivorous[36,37] and feed on marine aggregates[38], and/or (ii) chaetognaths feed on copepods, whose phenology coincides with the spring bloom and POC flux[29]. Nevertheless, further investigation and experiments are needed to better understand the role of these organisms in the carbon flux.

The summer period was also highlighted by the high abundance of the radiolarian order Chaunacanthida and its positive correlation with POC flux, indicating its essential role in carbon export in the Fram Strait. Chaunacanthida is an omnivorous particle feeder that can directly ballast marine aggregates. It can also contribute to carbon export by producing dense reproductive cysts that sink rapidly through the water column. Radiolarians are regularly detected in sediment traps[39–41] and have been shown to contribute between 3% and 22% of the total POC flux at 2000 m depth[41].

### Spatiotemporal changes of the benthic communities and pelagic-benthic coupling in relation to POC export

Though the Arctic benthic ecosystem is often considered stable, our 15-year record showed considerable spatial and temporal shifts of eukaryotic lineages in the surface sediment. No significant correlation was observed along the latitudinal gradient following the 2500 m isobath, supporting previous studies revealing no latitudinal trends in benthic bacterial and macrofaunal communities[42,43]. We expected differences in the benthic communities because there were different sea-ice conditions across latitudinally distributed stations. It, therefore, appears that lateral advection of the sinking matter reduced the regional differences and caused a more homogenized benthic ecosystem. However, when viewed from a longitudinal perspective, we could explain most of the spatial variability of the benthic subnetworks. These differences in sampling sites are primarily driven by (i) the bathymetric changes off Svalbard from HAUSGARTEN stations HG-I to HG-IX[17], and (ii) the seasonally ice-free stations in the West Spitsbergen Current (i.e., HG, N and S stations) and ice-covered stations in the East Greenland Current (i.e., EG stations)[3]. Based on the composition and spatial distribution of Sn_2, 4, and 5, we suggest that these benthic subnetworks show a sea-ice influence on the deep surface seafloor ecosystem. Although sea-ice diatoms contribute substantially to POC flux to 200 m water depth and high carbon export efficiency at the ice edge[8,29], we did not observe any correlation between sea-ice diatoms and benthic carbon. This might be a regional pattern, as Boetius et al.[7] observed widespread depositions of the *Melosira arctica* colonies on the central Arctic deep-seafloor. Two reasons could explain the lack of correlation between sea-ice algae and benthic carbon: (i) photosynthetic carbon assimilation by sea-ice algae is limited compared to their phytoplankton counterpart[44], limiting the amount of carbon exported and contained in our sediment cores, and (ii) the large ice diatom colonies such as *M. arctica* induce episodic carbon fluxes[7] that are temporally and spatially scarce and therefore difficult to sample. Furthermore, even if the shape and size of the sea-ice colonies are difficult for planktivores to graze or digest, they are an essential food source to the seafloor and are rapidly consumed by benthic megafauna[7,45]. However, the high abundance of sea-ice diatom ASV sequences in several benthic samples suggests that sea-ice enhances eukaryotic vertical connectivity, as recently observed for microbial communities[8].

The pelagic diatoms *Chaetoceros* sp. and *Thalassiosira* sp. significantly influenced the benthic carbon content. While both species are preserved as resting spores in the surface sediment[46,47], our study identified them as key species for carbon export. The contribution of benthic Sn_1 in the sediment dataset at the HG stations shows interannual variation with a decreasing trend from 2005 to 2008, reflecting

the shift in micro- and nanophytoplankton communities in the upper ocean during a warmwater anomaly in the eastern Fram Strait[17,48]. During this period, a higher contribution of Sn_6 indicated that Metazoa (i.e., Annelida and Nematoda) and dinoflagellates increased relative abundance. In 2008, when the warmwater anomaly was over, increased ice cover improved the development and export of diatom blooms in the catchment area[49], and the diatom-dominated Sn_1 replaced Sn_6 as the dominant contributor to the benthic community. While the literature provides limited information on sedimentation rates specific to sites within the Fram Strait, it is recognized that sedimentation rates can vary locally due to depth, geographic location (e.g., eastern vs. western Fram Strait), and proximity to the marginal ice zone. However, general observations indicate that sedimentation rates in this region are relatively low (i.e., 1–2 cm per 1000 years), and there is considerable sediment mixing due to bioturbation[50]. Although we aimed to minimize the temporal overlap by focusing on surface sediments- believed to retain a strong signal from recent pelagic blooms- and by conducting benthic samples after large sedimentation events (i.e., during the summer), we cannot entirely exclude the possibility of temporal overlap. Pelagic radiolarians, belonging to the order Chaunacanthida, showed a continuous positive correlation with POC flux and seafloor carbon content throughout the entire study period, highlighting the fundamental role of radiolarians in pelagic-benthic coupling[11,39–41].

### Ecological aspects and parasitism within the pelagic-benthic coupling

A high DNA abundance of parasites was observed in this study, corroborating previous findings observed in oligotrophic pelagic ecosystems[11] and on the seafloor[46]. However, their carbon association remains poorly understood.

Protist parasites are taxonomically diverse, able to modulate phytoplankton blooms[51–53], and essential players close to the sea-ice edge[23]. Among them, we identified taxa of the orders Cryomonadida (i.e., Protaspa lineages) and Syndiniales (i.e., group-I and II) and the genus *Pirsonia* as key lineages associated with the oceanic carbon cycle and subnetwork centrality. Cryomonadida taxa and *Pirsonia* sp. are known to be diatom-associated, feeding on diatom cytoplasm using pseudopodia that penetrate the shell[54]. The Syndiniales clades are associated with various hosts worldwide[53] and contribute to flux attenuation through remineralization[24]. Clades 1 and 2 of the Syndiniales group-I, the most abundant lineages in the sinking material and seafloor sediments, have been isolated from radiolarians and cercozoans[52,53]. Protist parasites induce physiological stress and occasionally cell lysis, which has been suggested to fuel the microbial loop by increasing the DOM and the POM release[46]. Here, our molecular tools and network analysis strongly indicate that the protist parasitic taxa, associated with key lineages of the biological carbon pump, such as the diatoms and radiolarians, contribute to the vertical export by being transported along the water column with their hosts. A high abundance of diatom parasites has also been observed in the Antarctic Peninsula sediment[46]. The authors suggested that parasites may infect *Chaetoceros* resting spores and adopt a life history strategy similar to their hosts by entering dormancy during unfavorable conditions, enhancing their vertical connectivity.

Protists are not the only group capable of infecting diatoms. Marine fungi, dominated by chytrid communities in the Arctic Ocean[55], are also susceptible to contaminating several diatom genera, such as *Chaetoceros* sp., *Navicula* sp., and *Pseudo-Nitzschia* sp., especially under the influence of sea-ice melt[56]. Chytrid infection is prevalent under light-stress conditions and for the genus *Chaetoceros* at low temperatures[57]. The Chytridiomycota class correlated well with the carbon content measured in our sediment cores. While chytrids can also exhibit saprotrophic feeding and the species level was not resolved, which limits our ability to classify their ecological roles, we suggest, as hypothesized by Hassett et al.[55], that the chytrids source in the sediment is related to the settling of individual algae cells and marine aggregates. Paradoxically, protists and chytrids have also been proposed to shunt the vertical organic matter fluxes by fueling the microbial loop (i.e., fungal shunt[58]) and the whole food web[59].

Despite their joint detection in the world's oceans and diverse hosts, understanding and incorporating the role of parasites into the biogeochemical models remains challenging[60]. Parasites can alter ecosystem functioning and redirect carbon fluxes, which has implications for the pelagic-benthic coupling. Graph alignment between pelagic and benthic co-occurrence networks suggests vertical connectivity of parasites following parasite-host interactions. Knowing that ecological aspects (e.g., parasitism, photoautotrophism, symbiosis) may drive pelagic-benthic coupling, we recommend additional effort should be invested in representing ecological and functional aspects of marine organisms rather than focusing on species taxonomy.

## Methods

### Collection and treatment of pelagic and benthic samples

The LTER observatory HAUSGARTEN is currently composed of 21 permanent sampling stations[17] (Fig. 1). Pelagic samples were collected with Kiel-type sediment traps (KUM trap type K/MT 234) moored in the water column between 179 m and 280 m water depths at the central HAUSGARTEN station (HG-IV; Fig. 1b) from 2000 to 2012. Before each deployment, the sampling cups on the sediment traps were filled with filtered seawater adjusted to a salinity of 40 PSU with NaCl and poisoned with mercury chloride (HgCl$_2$: final solution of 0.14%) to preserve the collected sinking material throughout the year. Moorings and sediment traps were exchanged annually. However, the traps sometimes did not collect sinking material due to electrical failures and deployment issues, which induced significant gaps in the time series. Methods and protocols used to estimate biogeochemical fluxes, swimmers' abundance, environmental variables, and climatic indices associated with each sample were published by Ramondenc et al.[29] and described in the supplementary material.

For benthic samples, the top five centimeters of the seafloor sediments were sampled using a TV-guided multiple corer (TV-MUC; corer ø 10 cm; length 50 cm) during the annual summer cruises to the HAUSGARTEN observatory between 2003 and 2018. In contrast to the pelagic sampling, nineteen HAUSGARTEN stations were investigated along longitudinal (i.e., HG-I to HG-IX) and latitudinal transects (i.e., S3 to N5), passing by stations regularly covered by sea-ice (i.e., EG-I to EG-IV) in the western Fram Strait. Although logistical constraints prevent us from obtaining a complete time series at all sampling sites, this provides a unique dataset to study spatiotemporal changes in the Fram Strait benthic ecosystem over 15 years. A detailed description of the methods used to analyze water content, biogenic sediment compounds, and estimated meiofauna abundance is provided by Soltwedel et al.[17,61] and described in the *Sample Treatment* section of the supplementary material. Among all benthic environmental variables used in this study, several core parameters- carbon content (Corg) and water content (H$_2$O)- were averaged, while particulate proteins, bacterial activities, phospholipid concentration (Lipids), and chlorophyll pigment equivalents (CPE) were summed over the top 5 cm of sediment. In contrast, other parameters, including chlorophyll a, bacterial activity, bacterial numbers, biomass, and mean bacterial biomass per Cell (MBC), were analyzed based on the first centimeter of sediment. To ensure that averaging and summing over the top 5 cm did not introduce bias compared to data from the first centimeter, we performed comparative analyses by examining the linear relationships between the measurements from these two sediment depths. Pearson correlation tests were conducted to assess the strength and direction of these relationships, and the results are presented in Fig. S3 of the supplementary file.

## Omics datasets

Sediment trap samples for eDNA analyses were wet split after manually removing zooplankton (swimmers and sinkers) with body sizes >0.5 mm. Molecular analyses were performed on 1/32 splits by filtration onto a 0.22 µm Sterivex-Filter (Millipore, Schwalbach, Germany). Filters were washed with sterile North Seawater (-50 ml). According to the manufacturer's protocol, genomic DNA was isolated from the samples with the PowerWater DNA Isolation Kit (Qiagen, Hilden, Germany). Benthic eDNA analyses were performed on the uppermost centimeter of sediment cores. Tipless syringes were used to collect three smaller subsamples from different TV-MUC cores and were stored at −20 °C until further processing. For DNA isolation, the top centimeter of the sediments was cut off from the syringe sample using a sterile scalpel. From each of the three subsamples, ~0.5 g sediment was pooled, while 0.25 g of this pool was subjected to further processing. Genomic DNA was isolated using the DNeasy PowerSoil Kit (Qiagen, Hilden, Germany) following the manufacturer's protocol. Negative controls are standard practice in molecular genetics, and we consistently incorporate them during extraction and PCR stages to monitor potential contamination. Agarose gels have been used to confirm the specific amplification of the 18S rDNA from our samples.

According to the manufacturer's protocol for measuring double-stranded, DNA concentrations of pelagic and benthic samples were determined using the Quantus Fluorometer (Promega, Germany). The resulting DNA extracts were stored at −80 °C until further analyses. For Illumina-Sequencing, a fragment of the 18S rDNA containing the hypervariable V4 region was amplified with the primer set 528iF (5′- GC GGTAATTCCAGCTCC-3′) and 938iR (5′-GGCAAATGCTTTCGC-3′). The library preparation, sequencing, and annotation of sequences were accomplished as described previously[62]. More details about the Illumina-Sequencing 18S rDNA and sequence analyses are provided in the supplementary material.

## Statistical analysis

To avoid the source of false-positive predicted association, singletons (i.e., 134 and 36 ASVs) were removed from the pelagic and benthic eDNA datasets. Each abundance matrix was filtered by keeping ASVs with 50 sequences in at least three samples.

Based on the sensitive mode with default parameters, pelagic and benthic network inference was performed using FlashWeave[63]. Commonly used for ecological network inference, FlashWeave infers all local associations between ASVs, before connecting their local dependencies to build a global network. This tool can include meta-variables (e.g., environmental variables) to report direct relationships between ASVs and meta-variables. As the FlashWeave workflow includes a centered log-ratio (CLR) transformation step[64], we used the filtered ASVs abundance matrices and associated meta-variables to construct the pelagic and benthic networks. Network stability, which refers to the robustness and resilience of the network, was assessed by analyzing node centrality and betweenness. These metrics helped identify keystone ASVs[65] that are essential for maintaining co-occurrence network connectivity and overall stability.

A WGCNA was applied independently on the pelagic and benthic CLR-transformed ASV abundance matrices to delineate subnetworks. This process is useful to (i) create a gene co-expression network, (ii) group genes with similar expression patterns in clusters (i.e., subnetworks), and (iii) correlate these subnetworks with environmental metadata to identify key drivers[66]. We applied the hierarchical clustering default method on the topological overlap measure calculated from a signed adjacency matrix with a soft thresholding power 6 to identify subnetworks. The association between subnetworks and meta-variables was then measured by the pairwise Pearson correlation coefficient between the subnetwork's eigengenes and environmental factors. A *p-value* threshold of 0.005 was applied to represent the significant

relationships. WGCNA computations and CLR transformation were performed using WGCNA[66] and phyloseq packages. As log transformation cannot be applied to zero values, a pseudo-count of min(relative abundance)/2 was applied to exact zero relative abundance. Before estimating diversity indices for the sediment core samples, rarefaction curves were generated from the sequencing data to assess whether the sampling depth was sufficient to capture the full diversity within the cores (Fig. S4). All samples reached an asymptote, confirming that the sequencing depth comprehensively represented the microbial diversity. Subsequently, diversity indices, including Shannon, Simpsons, and Chao, were calculated from benthic ASVs using the estimate_richness function from the phyloseq package.

A partial least squares (PLS) regression was performed on pelagic and benthic matrices using the R package pls[67]. This additional statistical analysis helps extract the ASV importance in projection (VIP) according to the response variable and identifies sequences playing a significant role in the PLS prediction. In the context of our analysis, higher VIP scores indicate ASV sequences that have a more substantial impact on the model's predictions, highlighting key drivers and providing insight into the most critical sequences for understanding observed patterns. Individual lineage analyses were also performed using a sparse PLS[68] approach, as described in the supplementary material.

To investigate pelagic-benthic coupling related to the carbon cycle, subnetworks significantly positively correlated with the carbon flux in the sediment traps, and carbon content in the sediment cores was compared using the graph-alignment technique L-GRAAL[69]. Initially used to compare protein networks, this method aligns nodes of two graphs based on topological and sequence properties. A consensus between both measures has been found according to the alignment quality scores (i.e., edge correctness and symmetric substructure scores) estimated by the alpha parameter that ranges from 0 (i.e., topological alignment) to 1 (i.e., sequence alignment). Hive-plot representation was chosen to highlight each ASV connection in its subnetworks and the ASVs' connections with neighbors' networks. Briefly, the nodes are ranked by relative abundance and duplicated on two axes to represent co-occurrence subnetwork structures. On each axis, nodes were divided into two groups, with internal and external groups representing the ubiquitous and dissimilar ASVs recorded in the pelagic and benthic ecosystems. The central edges represent the alignments between ASVs according to the alpha consensus parameter mentioned above. The node sizes highlighted the importance of the ASVs in both hiveplots, representing the degree centrality score.

The supplementary material provides an overview of the methods' limitations and uncertainties.

## Reporting summary

Further information on research design is available in the Nature Portfolio Reporting Summary linked to this article.

## Data availability

The raw sequence data generated in this study have been deposited in the European Nucleotide Archive (ENA) database under accession code PRJEB76183 for sediment cores and PRJEB74771 for sediment traps.

## Code availability

Additionally, the scripts for the pelagic and benthic network analyses are accessible at: https://github.com/sramondenc/NCOMMS-24-22914_Pelagic_Benthic_Coupling_Arctic/.

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

## Acknowledgements

We thank the captains and crews of RV Polarstern and RV Maria S. Merian for expeditions to the Fram Strait from 1999 to 2018, as well as the chief scientists. We also thank Swantje Ziemann and Kerstin Korte for their technical support in sequencing the eukaryotic microbial communities in this study. We greatly acknowledge the bioinformatics core facility of Nantes (BiRD, university of Nantes) and Alfred Wegener Institute (AWI), Helmholtz Center for Polar and Marine Research, for computational support. This work was conducted in the framework of the HGF Infrastructure Program FRAM of AWI. It was part of the Helmholtz Young Investigators Group PLANKTOSENS (VH-NG-500), the Helmholtz Exzellenznetzwerk POSY (The Polar System and its Effects on the Ocean Floor), the Deutsche Forschungsgemeinschaft/ Cluster of Excellence "The Ocean in the Earth System": EXC-2077-390741603, EU Projects SEA-Quester (101136480) and OceanICU (101083922), and the Helmholtz Program "Changing Earth—Sustaining our Future" supported by the Helmholtz Program-Orientated Funding (POF IV) to Topic 6 (Marine Life). SR was supported by funding from the internal AWI program "INternational Science Program for Integrative Research in Earth Systems" (INSPIRES).

## Author contributions

S.R. conceived the study and wrote the manuscript with contributions from D.E., K.M., E.N., D.P., M.H.I., C.H., and T.S. S.R. performed data analysis and plotted figures under D.E.'s supervision. K.M. provided eDNA data. All authors revised, edited, and approved the manuscript.

## Funding

## Competing interests

The authors declare no competing interests.
