## [Peer Review File · Nature Communications]

Unveiling pelagic-benthic coupling associated with the biological carbon pump in the Fram Strait (Arctic Ocean)

Corresponding Author: Dr Simon Ramondenc

Version 0:

Reviewer comments:

Reviewer #1

(Remarks to the Author)

This is a very interesting study that many researchers have thought about conducting, but didn't have the technological means and long timeseries data to complete. I am glad to see the field advancing to a point where it is now possible to compare pelagic and benthic genomic coupling via POC export. The findings of the importance of marine parasites to POC export or recycling is very interesting. This manuscript should be published after addressing minor revisions and concerns.

Do you have any information on the sedimentation rate across the different depths at which you took the cores? From Figure 1, it looks like you collected benthic sediment samples from ~1000-5000 meters across your study site. Would using the upper 1 cm (or 5 cm? it is unclear between the description in the Introduction and the description in the Methods what depth of sediment you sequenced) be essentially sampling POC exported over varying timescales. For example, the top 1 cm of sediment may be reflective of the most recent year of export from the euphotic zone if export rates are high, or it may be the accumulation of the most recent 100 years of export if export rate is low and how deep your benthos is. So there may be a mismatch with your temporal analysis, within the sediment itself across space, but also a temporal mismatch between the benthic community and the overlying pelagic community.

Were there contemporaneous zooplankton net tows or water column eDNA taken from the CTD that you can compare the sediment trap and benthic DNA to?

In the results section, you did not discuss any results from the second half of Fig 4c with regards to latitude - were there any interesting findings?

It seems like the latitudinal samples were taken further north from where the pelagic sediment trap was. Do you see a separation in benthic community structure as you move north?

For the longitudinal samples, you are not only moving further from where the pelagic sediment trap samples are being collected, but are moving deeper (or shallower) for your benthic samples. Are you finding that overall distance from the trap affects the community in reference to the pelagic community from the trap? Are you finding that the depth of the benthic samples affects the community structure? I would expect the deeper samples to look less like the pelagic samples as there would have been more flux attenuation and less coupling between the surface and benthic communities.

Did you find any species in the benthic ecosystem not found in the pelagic ecosystem? That would be interesting to note and explore.

I am not sure if the authors are already aware, but there is a pre-print publication showing similar results in the Sargasso Sea with regards to the negative correlation between Syndiniales and POC export.
doi: <https://doi.org/10.1101/2023.06.29.547083>

-Line 46: I believe you mean "dawn" and not "drawn"

-Line 47: please elaborate on the "processes" you are referring to

- Line 83: you state here that you used the "upper centimeter" for the sediment cores, but in the methods, you discuss sampling the top 5 cm of sediment. Did you subsample the cores? Please elaborate/clarify. Additionally, it would be helpful to state the depth range of your sediment cores in your analysis.
- Line 92: missing the word "Gene" in the WGCNA
- Line 94: please define what ASV stands for (amplicon sequence variants). It is good to see you analyzed the genomic sequences using ASVs instead of OTUs.
- Line 85-96: the sentence "Five subnetworks were dominant..." is confusing. Please elaborate.
- It is confusing you discussed Figure 3a, and skipped to Figure 4 and then back to Figure 3b several paragraphs later. I would suggest including Fig 3a into Fig 2 and moving Fig 3b into Fig 4, and make the comparison diagram in Fig 3c the new Fig 5, and push all the other figures down by one.
- Line 152: I believe you mean "Venn", not "Veen"
- Lines 167-168: you mention the relationship between Sn2, Sn4, and Sn5 to ice cover, did you compare these sequences and communities with sea ice cover duration, extent, onset day or retreat day for that year? It looks like you compared the pelagic sequences to environmental parameters such as sea ice, but did not do this comparison for the benthic sequences?
- Line 222: Did you see an increase in Chaetognatha during this same time period from 2007 to 2012 in both the benthos and pelagic samples? Did some environmental change occur from 2007 to 2012 that increased their abundance?
- Lines 250-252: It would be helpful to have this description in the introduction and in your Figure 1 caption to help readers understand why you sampled these different regions and how environmentally different they are.
- Line 314: I believe you meant "diverse" and not "divers"
- Lines 330-332: You state that more details on methods can be found in the citations and supplementary materials, but it would be beneficial to explicitly state here what formula of chemicals you filled the sediment trap bottles with, as different types of preservatives can impact DNA preservation and extraction.
- Line 334: In the introduction, it states you performed genomic analysis on the "upper centimeter" of sediment core samples (Line 83). Did you subsample the 5cm down to 1cm? It is unclear from your description.
- Line 347: you mention swimmers here, but in Figure 2, there is a mention of sinkers and swimmers. Please clarify the differences.
- Line 379: missing the word "Gene" in WGCNA
- Figure 1 caption: Please add more details about the abbreviations shown in the map. What does HG stand for? N? EG? S3? What are the red points (sampling stations). And the abbreviations for the currents.
- Figure 2A: What is meant by Total in the biogeochemical fluxes section? Is this total particulate organic matter or total dry weight of organic and inorganic particles? Does this include dissolved biogeochemical fluxes? It is interesting that Sn11 is related to POC flux and Total Flux, but Sn3 is only correlated with POC flux. Please address it in the figure or in the figure caption and in the text if this was an important parameter.
- What is meant by "u ice" and "v ice"? I believe "u wind" and "v wind" are the velocities in the u and v direction of wind (or current), but I am unfamiliar with this in reference to ice. Is this the drifting velocity of sea ice over top the trap? Does ice drifting speed impact POC export, as opposed to ice concentration or total ice cover or length of the ice season? Please address it in the figure or in the figure caption and in the text if this was an important parameter.
- What do you mean by swimmer and sinker fluxes? Are these zooplankton counts from overlaying net tows? From reading the supplementary materials and cited references, it seems the sinkers naturally sank into the trap as opposed to the swimmers that accidentally swam in and died - are the sinkers counted as part of the POC flux and Total flux? Please address it in the figure or in the figure caption and in the text if this was an important parameter.
- Figure 2B and 2C: Can you label the circles SN 1, Sn 2, Sn 3, instead of just 1, 2, 3? It was confusing at first to figure out what the numbers referred to. Likewise, can you make a key/legend for 2C relating each color to each Sn?
- Figure 4A: What is meant by AFDW? Please address it in the figure or in the figure caption and in the text if this was an important parameter.
- Figure 5 caption: can you add details on how to read/interpret the plot in 5b? Why are the taxa in the order that they are on each side? What does the width of the lines indicate?
- Supplementary Figures: can you describe the waves in part B of each figure, what the colors stand for and how to interpret the plot?

(Remarks on code availability)

The code is not currently available on github.

Reviewer #2

(Remarks to the Author)

This manuscript presents a unique time-series of eDNA from sediment trap data collected in the Fram Strait over more than a decade, and coupled with benthic sediment samples. It provides important information on carbon fluxes and benthic-pelagic coupling in this well-studied waterway between the Arctic Ocean and the Greenland Sea. The co-occurrence network analyses are an effective way to reduce the amount of data generated by metabarcoding and the comparison of ASV and sub-network occurrences with specific environmental parameters gives a useful overview of the data and their ecological significance.

I find it particularly interesting that this study highlights the importance of parasites, a group that has so far been overlooked in climate change ecology studies.

The manuscript is very clear and well written but it is rather concise, and some important information are lacking, in my opinion, particularly when it comes to the methods and the rationale behind the sampling design for the study.

My main points of concern are:

1 - The authors claim that they have a 15-year long time series of eDNA sequences from sediment samples. These samples were collected with multicorers and represent the upper 1cm of sediments, that have been mixed and from which DNA was extracted. I was puzzled to find no mention of sedimentation rates and how they vary between sampling sites, no data on age control for these sediments, and no discussion on the fact that one can expect a large degree of temporal overlap between some samples, as sedimentation rates are generally low in this region. It is therefore perhaps not surprising that the authors find the benthic networks to be more "self-connected" than the pelagic networks.

2 - Choice of DNA marker. In the introduction there is reference to the Tara Oceans initiative, however, the study uses V4 (as opposed to V9) and there is little to no discussion on how the choice of marker affects the results. It is not clear why a single marker was chosen as opposed to multiple markers, and why V4? Point 2 can be addressed by simply explaining the choice of marker and discussing the implications and potential biases in taxonomic coverage.

Regarding point 1, this is a major concern, as analysing a bulk 1cm-thick sample of seafloor sediments collected over 15 years cannot be equalled with having a 15-year long time series, and this needs to be corrected, supported by sedimentation rate data, and carefully discussed.

(Remarks on code availability)

Reviewer #3

(Remarks to the Author)

The manuscript by Ramondec and co-authors is describing an impressive long-term eDNA dataset in which they combine sediment trap data and seafloor sediments to understand the sources of the particulate organic carbon fluxes arriving at the seafloor on both a spatial (seasonal and interannual) and temporal scale. While DNA metabarcoding and correlation network approaches themselves are not new (for example Djurhuus et al. 2020: doi: 10.1038/s41467-019-14105-1) and studies coupling eDNA from sediment traps and sediments from the seafloor exist, the novelty lies rather in the timeframe of data collection and the association of the DNA with POC fluxes. I think the dataset is of significance for the field because long-term studies covering more than 10 years are rare and the results very interesting. Therefore, the manuscript is worth considering, but in its current form not mature enough. I have several concerns and additional line-by-line comments listed below.

Major concerns:

1. My main concern is the lack of reporting of any negative controls (extractions and PCRs) and I wonder if they were just not reported or if they were not done. eDNA is susceptible to contamination and reporting on negative controls is a standard quality control procedure. If they were just not reported, the authors should briefly explain what was found in them and how they treated the data based on that knowledge.

2. Also, PCR replicates were not reported. There is considerable literature out there showing that results from different replicates can vary substantially, which could affect the outcome and robustness of this study. Also, I could not find any specifications of amplification conditions (polymerase, cycle numbers, annealing temperature...). Please, add the information, at least in the supplement, as those all factor into the final datasets.

3. There is no real introduction of the methods and some terms are used undefined (I mention them specifically in the line-by-line comments).

4. Overall, I am missing details to the strength of the correlations. Why was the Pearson method used? It requires strict linear responses, but what if relationships are non-linearly? Is it sensitive enough to answer your questions? Is the proposed method (WGCNA) appropriate for the analysis? Is your dataset sparse and if yes, how does this method deal with sparsity? Is POC correlated with other environmental factors and how is this affecting the results? If L-GRAAL is a novel method applied to this kind of data, it requires a better introduction on how the algorithm works.

5. Based on the low to very low correlation coefficients between 0.1-0.3 for chaetognaths and *Micromonas polaris* and <0.4 for *Melosira arctica*, highlighting them as major contributors to carbon export is not convincing. If they were among the most abundant reads, it would be more convincing.

Minor concerns:

1. In their introduction, the authors mentioned that they “Aim to understand how the biological carbon pump affects diversity and vice versa” but I feel this overarching aim is not well reflected in the manuscript. Diversity is only mentioned on the side, and it is unclear how it was estimated methodologically.
2. I did not find a statement that the data will be deposited in a public repository after publication. But maybe this is not part of the reviewer version.

Line-by-line comments:

Abstract:

L21: I think instead of driver the word components should be used. The taxa are rather components of the vertical carbon flux, while downwelling for example would be a driver.

L27: I don't think that you show that parasites induce export mechanisms, you rather mentioned sedimentation with hosts in the discussion and that resting spores can be infected, too. Please rephrase.

L39: coupling pelagic and benthic diversity – remove microbial

L55-58: Can you please cite a paper here?

L83: What is the sedimentation rate at the sampling site and how much time is represented by the upper 1 cm?

L84: There is no previous introduction that you are going to use network analysis. So the part where you explain that you isolate clusters related to the carbon cycle comes very abruptly and it confused me. I thought you had also analyzed metagenomic data and inferred gene clusters involved in the process. Can you please introduce the method briefly?

Results:

L92: “weighted gene co-expression network analysis”

L91-93: Which algorithm was used to identify the subnetworks?

L98: How do you know they are directly correlated to carbon export and not indirectly via a third variable? Maybe write instead that they were correlated with POC.

L101: Just remove “Its relative abundance showed that”, because this does not make sense.

L105: Instead of carbon export, please write POC flux.

L118: You refer to Fig. 2a, but I don't see any microscope count according to the figure. Is it referring to the “environmental data” category? If so, please specify this in the caption and refer to the supplement.

L121: How do you define network stability and where do you show this? Most readers are probably not experts in network analysis. I suggest you define the network properties to make it more comprehensible, e.g. explain the relevance of network stability.

L121-125: There are many measures of centrality Which was applied here? Degree centrality, betweenness,...? Please specify it here and in the figure caption. Are key lineages those with taxonomic name tag in Fig. 3a/b. How did you decide which bubbles you label and which not? I don't understand why nodes with coefficients >0.5 are not labelled and while some with coefficients of 0.1-0.3 are and it seems to me a bit like cherry picking. The correlation coefficients of several ASVs highlighted in the text (*chaetognaths*, *M. arctica* and *M. polaris*) are not convincing enough to classify them as major contributors to carbon export and I think they should be removed from the text.

L127: “Benthic” is a bit confusing in that sense, because it contains also deposited organic matter/DNA from pelagic species. Maybe call it “sediment dataset” or define what you mean in the first sentence.

L132: Where do these variables come from? Which diversity indices were used? Was a rarefaction applied to account for differences in read counts between samples? It would be good if you can refer to the specific sections in the supplement.

L139: remove “increasing” – is it really increasing or just higher? Can you show the data?

L145: A weak contribution to what?

L148: “Irrespective of their abundance, several Sn₁ lineages contributed strongly to sediment carbon context.” Can you please specify what you mean? Right now, I am not convinced by that statement. You do not factor in copy number differences or any potential biases from PCRs.

L152: Venn

L152-154: Why not remain at ASV level? I think family level is too broad here and only the ASV/species level can support your suggestion of vertical coupling. The order Chytridiomycotina contains probably more saprotrophs than parasites. Hence, they might also just feed on the organic matter/the carbon sources. And have you checked whether parasitic taxa are correlated with potential host taxa?

L157-159: Many diatoms referred to here as sea ice diatoms are in fact not truly sympagic and can occur in cold waters with or without sea ice, for example *F. cylindrus* (check Oksman et al. 2019; doi: 10.1016/j.marmicro.2019.02.002). And *Navicula* and *Nitzschia* are not per se sea ice associated. Maybe rephrase it to something like: “Various taxa often detected in sea ice, ...”

L160: What does “large diversity” mean? Please give number of ASVs as this is interesting. Are they phylogenetically close?

L163: typo: Corethron – also, I did not know this as a sea ice species. Can you cite this please?

L172: Introduce the graph alignment analysis – I don't understand how it infers the same ecological roles in the 2 networks within the community structure? A diatom in the water column is photosynthesizing and part of a community, but a diatom at 2000m depth is not fulfilling that role anymore and serves rather as food. Can you clarify please what you mean? What is the alpha value? Again, what measure of centrality?

L179-180: I don't really understand what you mean by that sentence. Also, please define keystone ASVs.

L180-181: 103 out of how many?

L182-184: I do not follow the reasoning for that. All ASVs that are found in the sediment traps and in the seafloor sediments

should suggest export. But how do you infer sedimentation as a distinct process for that?

L184-185: Why are you now jumping to phylum level? Is this adequate and really providing a holistic view?

Discussion

L215: typo: Phaeocystis

L225: From the figures, it looks like those taxa have a correlation ~ 0.3 , so rather weak and not strong. How do you conclude that they are key species for carbon export?

L250: I did not read anything about the bathymetry being a driver in the results part. Maybe I overlooked it, but if not, I suggest you highlight it or write a sentence about it in the results.

L255: is the high carbon export efficiency inferred from the literature or from the data?

L272: is it pelagic Sn_1?

3.3 Ecological aspects and parasitism within the pelagic-benthic coupling

Overall, I find the information presented here interesting and sound, but I am not convinced that the chytrids are parasitic. It could be, but if they are taxonomically resolved at such a high level, I ask myself how you get to this conclusion. Can you provide more information why you attribute parasitism and not saprotrophic feeding?

L 314: typos in sentence

Methods:

WGCNA: The variables have different scales, how were they treated/transformed?

Figures:

Fig. 2+4: The captions are very unspecific and abbreviations are not explained. What is u and v? where are the environmental variables taken from? Explain abbreviations. In the plots it is *S. elegans*, but in the text it's *P. elegans*.

Fig. 3: What is the VIP score?

After Fig. 5, there are figures without any label and captions. The heatmaps are not readable.

Supplement:

Which algorithms were used for denoising and chimera removal?

Taxonomic assignments were carried out with which program?

L85: How many (%) of ASVs are unknown?

L90: How is this influenced?

Code availability: The URL does not work.

(Remarks on code availability)

The URL did not work, therefore, I could not review the code.

Version 1:

Reviewer comments:

Reviewer #1

(Remarks to the Author)

I am mostly satisfied with the authors' revisions.

As per my original comment regarding switching back and forth between discussing Figure 3 and Figure 4 in the text, if the authors do not want to combine figures, I suggest making Figure 3B and 3C a separate figure - they can be combined into a new Figure 5. It is traditional to discuss all panels of a figure in the text before moving on to the next figure, so switching back and forth between Figure 3 and Figure 4 is confusing to readers.

Additionally, the github url provided for access to the code does not work: https://github.com/sramondenc/Pelagic-benthic_coupling

(Remarks on code availability)

Reviewer #2

(Remarks to the Author)

I think the authors have done a superb job at addressing the comments from all reviewers and the manuscript is now much improved. I have only one comment regarding the use of V4. The limitations with the use of a single marker and particularly V4 should be addressed in more detail, particularly in light of recent work comparing the performance of multiple 18S markers (see <https://doi.org/10.1002/edn3.580>).

(Remarks on code availability)

Reviewer #3

(Remarks to the Author)

Dear Dr. Simon Ramondenc and co-authors. Thank you for considering the comments I made in your revised version. In my opinion the manuscript is now more clearly written and contains a bit more background. I have just a few comments left, that in my opinion still need to be addressed:

1. Negative controls: In my personal experience, even negative controls without visible bands in the gels can reveal some contamination after sequencing. Since it cannot be changed now, I strongly suggest that the authors at least add a sentence into their Material & methods part about their application of negative controls. Not reporting on this, particularly in a high impact journal, can give the false impression that negative controls are not necessary.
2. Replicates: In my opinion, writing that replicates are omitted based on one replicate in one study is not a good signal to the research community. The abundant taxa are usually consistent but not the rare ones. Please rephrase to make clear that replication is actually encouraged and that it would most likely strengthen your signals.
3. Sedimentation rates: I am still puzzled by this. If there is such a low sedimentation rate, and if we consider bioturbation, would you not expect the samples to contain significant temporal overlap? I think this still needs to be addressed clearer in the manuscript.
4. I asked for a better introduction of the network metrics. I think terms such as 'keystone ASV' should be introduced at least a bit the first time they appear in the manuscript, not buried in the methods.
5. Chimera removal is typically carried out with a dedicated algorithm (e.g. usearch / vsearch). I am not sure that your approach compensates for all chimeras.
6. Fig. 2: There is still no information on what u or v means in the caption. Please, add this.
7. The code is still not free for review. Just remember to publish it after publication.

(Remarks on code availability)

The code is still not free for review, which is why I could not review it.

Reviewers comments on «Unveiling the pelagic-benthic coupling associated with the biological carbon pump in the Fram Strait (Arctic Ocean)» Reference: NCOMMS-24-22914

This is a re-submission

Reviewer's comments in bold – Responses in black normal –

Text added to the manuscript in blue -

Reviewer(s) comments

Reviewer #1

This is a very interesting study that many researchers have thought about conducting, but didn't have the technological means and long time series data to complete. I am glad to see the field advancing to a point where it is now possible to compare pelagic and benthic genomic coupling via POC export. The findings of the importance of marine parasites to POC export or recycling is very interesting. This manuscript should be published after addressing minor revisions and concerns.

We thank reviewer#1 for the support and positive feedback on our manuscript. We are pleased to hear that you find our study interesting and timely, especially in light of the technological advancements and long-term data that have made this research possible. We also appreciate your acknowledgment of the importance of our findings regarding marine parasites and their role in POC export and recycling. Your encouraging words reinforce the value of this research and its contribution to the field. We carefully addressed the revisions and concerns you have raised. We provided a point-by-point response to the minor revisions and incorporate the necessary changes into the manuscript.

Do you have any information on the sedimentation rate across the different depths at which you took the cores? From Figure 1, it looks like you collected benthic sediment samples from ~1000-5000 meters across your study site. Would using the upper 1 cm (or 5 cm? it is unclear between the description in the Introduction and the description in the Methods what depth of sediment you sequenced) be essentially sampling POC exported over varying timescales. For example, the top 1 cm of sediment may be reflective of the most recent year of export from the euphotic zone if export rates are high, or it may be the accumulation of the most recent 100 years of export if export rate is low and how deep your benthos is. So there may be a mismatch with your temporal analysis, within the sediment itself across space, but also a temporal mismatch between the benthic community and the overlying pelagic community.

Thank you for your insightful questions and observations. Regarding the sedimentation rates across different depths where we collected the cores, it is essential to note that mean sedimentation rates in the Fram Strait are relatively low, averaging 1-2 cm per 1000 years. In the eastern parts of the Fram Strait, sedimentation rates can rise to approximately 30 cm per 1000 years, based on data from 1500 meters depth at 78°50'N. Despite these higher rates in some regions, overall sediment accumulation remains minimal and should not significantly impact our findings between the different water depths. The sediment cores that we sampled span depths from 1000 to 5000 meters. Given the low sedimentation rates, bioturbation is a more significant factor influencing the upper sediment layers compared to new sediment deposition¹. This continuous bioturbation perturbs the upper sediment layers, potentially mixing material over timescales that might obscure short-term variations in particulate organic carbon (POC) export. This broad temporal integration means there could indeed be a temporal mismatch in our analysis, both within the sediment itself, across different locations, and between the benthic and pelagic communities. The benthic community structure and the POC sampled might not directly correspond to recent surface productivity but rather reflect a historical accumulation influenced by slow sedimentation and active bioturbation. We appreciate your input, highlighting the complex dynamics between sedimentation, bioturbation, and POC export. We have addressed these points thoroughly in the *Method* section (L.364-L.375).

Were there contemporaneous zooplankton net tows or water column eDNA taken from the CTD that you can compare the sediment trap and benthic DNA to?

Thank you for your question regarding contemporaneous zooplankton net tows and water column eDNA. Yes, there have been continuous zooplankton net tows conducted in the Fram Strait since 2010 by Barbara Niehoff's team, and eDNA data from CTD casts have been collected since 2009. However, directly comparing these datasets to our sediment trap and benthic DNA data presents significant challenges due to differences in instruments and timescale observations. Zooplankton net tows and water column eDNA samples collected using CTDs provide high-resolution, short-term snapshots of the pelagic community at specific times and locations. These data are valuable for understanding the water column's immediate and recent biological activity. On the other hand, sediment traps and benthic sediment samples integrate signals over much longer periods. Sediment traps collect particulate matter settling through the water column over weeks to months and integrate particles with different settling velocities and ages, while benthic sediment samples represent accumulation over years to centuries due to low sedimentation rates and strong bioturbation effects. This results in a much coarser temporal resolution for the sediment and benthic DNA data. Because of these differences, direct comparisons between the datasets are challenging. The temporal and spatial resolution of the eDNA from CTD data does not align with the integrated, long-term nature of the sediment and benthic data.

In the results section, you did not discuss any results from the second half of Fig 4c with regards to latitude - were there any interesting findings?

In our analysis, we focused on identifying significant relationships and patterns. According to Figure 4a, we found no significant correlation with latitude. As a result, no subnetwork was driven by the latitudinal gradient. This lack of a significant latitudinal effect explains why we did not discuss findings related to latitude in the second half of Figure 4c. Our discussion centered on the most relevant and impactful results, and the absence of a latitudinal gradient effect meant it did not warrant further elaboration in the results section. We appreciate your attention to detail and will provide additional clarification if needed. Still, according to our analyses, latitude alone did not impact the vertical connectivity or the community structure, we have clarified this in the manuscript (L.252-L.254).

It seems like the latitudinal samples were taken further north from where the pelagic sediment trap was. Do you see a separation in benthic community structure as you move north?

As mentioned above, our analysis did not identify any benthic community structure driven by the latitudinal gradient. Despite the latitudinal samples being taken further north from where the pelagic sediment trap was located, we did not observe a separation in the benthic community structure as we moved north. Our results indicate that latitude did not significantly influence the benthic community composition in this study area.

For the longitudinal samples, you are not only moving further from where the pelagic sediment trap samples are being collected, but are moving deeper (or shallower) for your benthic samples. Are you finding that overall distance from the trap affects the community in reference to the pelagic community from the trap? Are you finding that the depth of the benthic samples affects the community structure? I would expect the deeper samples to look less like the pelagic samples as there would have been more flux attenuation and less coupling between the surface and benthic communities.

We did not perform a specific analysis comparing the benthic community structure to the pelagic community from the sediment trap based on sample location. However, as shown in **Figure 4a**, some benthic subnetworks are significantly correlated with the longitude or depth. As we mentioned in the main text, the longitudinal gradient is related to both bathymetric and sea-ice variations among the HAUSGARTEN stations. Specifically, the differences observed across the stations from HG-I to HG-

IX off Svalbard can be attributed to bathymetric changes. Your expectation that deeper samples would show less similarity to pelagic samples due to more flux attenuation and reduced coupling between surface and benthic communities is plausible. Our findings reinforce the importance of depth, indicating that it significantly influences the benthic community structure, likely due to factors such as flux attenuation and varying environmental conditions with increasing depth.

Did you find any species in the benthic ecosystem not found in the pelagic ecosystem? That would be interesting to note and explore.

We observed sequences in the benthic ecosystem that were not found in the pelagic ecosystem. These are presented in the caption for Figure 5a. The external nodes in Figure 5a showed ASVs in the pelagic or benthic subnetworks that correlated to the carbon cycle. This finding highlights the presence of distinct benthic species not part of the pelagic community, possibly due to the unique environmental conditions and ecological niches in the benthic ecosystem.

I am not sure if the authors are already aware, but there is a pre-print publication showing similar results in the Sargasso Sea with regards to the negative correlation between Syndiniales and POC export.

doi: <https://doi.org/10.1101/2023.06.29.547083>

Thank you for bringing this to our attention. We read the publication regarding the negative correlation between Syndiniales and POC export in the Sargasso Sea and included it as a reference in our manuscript as follows (L.209-L.210): “These results align with a recent publication indicating that Syndiniales play a significant role in flux attenuation through remineralization²”.

-Line 46: I believe you mean "dawn" and not "drawn"

You are correct; “dawn” is the appropriate term in this context and replaced in the revised manuscript.

-Line 47: please elaborate on the "processes" you are referring to

We are specifically referring to biogeochemical processes. These include the chemical, biological, and physical interactions that influence the cycling of elements and compounds in the marine environment, such as nutrient cycling, organic matter decomposition, and primary production. We added the term “biogeochemical” to this sentence, such as (L.46-L.49) “The dawn of improved and more affordable sequencing technologies has enabled researchers to discover formerly unknown organisms, and gain insights into critical processes such as carbon and nitrogen cycles¹¹, microbial interactions¹², and evolutionary adaptations in marine ecosystems.”.

-Line 83: you state here that you used the "upper centimeter" for the sediment cores, but in the methods, you discuss sampling the top 5 cm of sediment. Did you subsample the cores? Please elaborate/clarify. Additionally, it would be helpful to state the depth range of your sediment cores in your analysis.

We appreciate Reviewer #1’s valuable feedback and for pointing out areas where further clarification was needed in our manuscript. In response, we would like to clarify that several core parameters (carbon content, water content H₂O, particulate proteins, bacterial activities, phospholipid concentration Lipids, chlorophyll pigment equivalents CPE, and ash-free dry weights AFDW) were averaged or summed over the top 5 cm of sediment. Meanwhile, other parameters (chlorophyll a, bacterial number, and biomass, mean bacterial biomass per Cell MBC) were analyzed based solely on the first centimeter of sediment. To ensure that using the top 5 cm did not introduce bias compared to data from the first centimeter, we performed an additional comparative analysis by examining the linear relationships between these measurements. For validation sake, Pearson correlation tests were conducted to quantify the strength and direction of these relationships. As illustrated in the figure

below (**Figure 1**), a significant correlation was observed between the data from the top 5 cm and the first centimeter for most parameters, except for AFDW. This last parameter was, therefore, excluded from the final analysis and the revised manuscript. This information has been incorporated into the revised manuscript, along with the new figure in the supplementary materials providing an overview of the correlation metrics (*i.e.*, R and p-value). Additional details have been added to the *Methods* section (L.358-L.368) to explain the sample treatment and the depth range used in the correlative analyses “Among all benthic environmental variables used in this study, several core parameters—carbon content (Corg) and water content (H₂O)—were averaged, while particulate proteins, bacterial activities, phospholipid concentration (Lipids) and Chlorophyll pigment equivalents (CPE) were summed over the top 5 cm of sediment. In contrast, other parameters, including chlorophyll a, bacterial activity, bacterial numbers, and biomass, and mean bacterial biomass per Cell (MBC), were analyzed based on the first centimeter of sediment. To ensure that averaging over the top 5 cm did not introduce bias compared to data from the first centimeter, we performed comparative analyses by examining the linear relationships between the measurements from these two sediment depths. Pearson correlation tests were conducted to assess the strength and direction of these relationships, and the results are presented in **Figure S3** of the supplementary file.”. We have also revised Figure 4 by removing the AFDW parameters and highlighting the core parameters averaged over the top 5 cm of the sediment core with an asterisk.

Figure 1. Linear regression analysis of core parameters (carbon content, water content H_2O , particulate proteins PROT, bacterial activities FDA, phospholipid concentration Lipids, Chlorophyll pigment equivalents CPE and ash-free dry weights AFDW) comparing data from the top 5 cm versus the first 1 cm of sediment. The *p*-values represent the Pearson correlation test results, indicating the correlations' statistical significance. R^2 values denote the correlation coefficients, reflecting the strength and direction of the relationships between the two depths for each core parameter.

-Line 92: missing the word "Gene" in the WGCNA

You are correct that the word "Gene" is missing in the description of WGCNA. We corrected this to read "Weighted Gene Co-expression Network Analysis (WGCNA)" in the manuscript.

-Line 94: please define what ASV stands for (amplicon sequence variants). It is good to see you analyzed the genomic sequences using ASVs instead of OTUs.

The acronym ASV is now defined in the revised manuscript.

-Line 85-96: the sentence "Five subnetworks were dominant..." is confusing. Please elaborate.

We revised this sentence to clarify our meaning, such as (L.98-L.100) "Five subnetworks showed high ASV abundance (Sn_3, Sn_4, Sn_9, Sn_10, and Sn_11) with 21.6%, 15.0%, 9.3%, 18.6%, and 18.3% of the total pelagic sequences.".

-It is confusing you discussed Figure 3a, and skipped to Figure 4 and then back to Figure 3b several paragraphs later. I would suggest including Fig 3a into Fig 2 and moving Fig 3b into Fig 4, and make the comparison diagram in Fig 3c the new Fig 5, and push all the other figures down by one.

We understand the concern about the flow and sequence of the figures. However, incorporating additional subplots into figures 2 and 4 could lead to further complexity and confusion due to the increased information density in these figures. Figures 2 and 4 are intended to provide an overview of the pelagic and benthic network analysis, respectively, while figure 3 focuses specifically on the POC flux and carbon content. By keeping figure 3 separate, we aim to ensure that the detailed POC flux and carbon content analyses are presented without overcrowding the overview figures. This separation helps maintain clarity and allows for a more focused discussion of each aspect of the data.

-Line 152: I believe you mean "Venn", not "Veen"

We corrected this in the revised manuscript.

-Lines 167-168: you mention the relationship between Sn2, Sn4, and Sn5 to ice cover, did you compare these sequences and communities with sea ice cover duration, extent, onset day or retreat day for that year? It looks like you compared the pelagic sequences to environmental parameters such as sea ice, but did not do this comparison for the benthic sequences?

Our study did not correlate ice conditions with the benthic sequences. This is because, in the ocean's hydrodynamics, the distance from the sea-ice edge becomes less relevant at the depths where our benthic samples were collected. Given that our benthic samples were taken at depths of several thousand meters and that the sedimentation rates are low, factors like sea-ice cover duration, extent, onset, or retreat day are unlikely to directly influence the benthic communities in the same way they affect pelagic sequences. However, our focus on comparing pelagic sequences with environmental parameters, including sea-ice, reflects their more immediate and direct interaction with surface processes.

-Line 222: Did you see an increase in Chaetognatha during this same time period from 2007 to 2012 in both the benthos and pelagic samples? Did some environmental change occur from 2007 to 2012 that increased their abundance?

We observed the highest abundance of Chaetognatha in the pelagic samples between 2007 and 2012. In contrast, no Chaetognatha sequences were detected in the benthic samples, indicating that the increase in Chaetognatha was confined to the pelagic environment and did not extend to the benthic community. Regarding environmental changes, our previous study³ identified a significant correlation between Chaetognatha counts and factors such as ice-edge distance and POC flux. The current research suggests that POC export and chlorophyll a concentration might be the key environmental factors influencing the increased pelagic Chaetognatha abundance. Further research is needed to explore these environmental changes in more detail to better understand the dynamics affecting Chaetognatha populations.

-Lines 250-252: It would be helpful to have this description in the introduction and in your Figure 1 caption to help readers understand why you sampled these different regions and how environmentally different they are.

We appreciate the importance of providing context for the sampling regions. We want to note that the specific environmental conditions driving our choice of sampling sites are already mentioned in the introduction (L.56-L.71). We have provided detailed the hydrographic dynamics of the Fram Strait, including the influence of the West Spitsbergen and East Greenland currents and the associated sea-ice variations that characterize the HAUSGARTEN stations. We have now included the bathymetric gradient in the text as an essential environmental parameter. This context helps underscore the sampling regions' ecological relevance and environmental diversity.

-Line 314: I believe you meant "diverse" and not "divers"

You are correct; the appropriate term should be "diverse" rather than "divers." We have corrected the wording issue in the revised manuscript.

-Lines 330-332: You state that more details on methods can be found in the citations and supplementary materials, but it would be beneficial to explicitly state here what formula of chemicals you filled the sediment trap bottles with, as different types of preservatives can impact DNA preservation and extraction.

We have included the details regarding the preservation method in the supplementary materials. To clarify, before each deployment, the sampling cups on the sediment traps were filled with filtered seawater adjusted to a salinity of 40 PSU with NaCl and poisoned with mercury chloride (HgCl₂: final solution of 0.14%) to preserve the collected material during deployment and after recovery. We understand the importance of specifying the preservatives used, as they can indeed impact DNA preservation and extraction, and we appreciate your suggestion to make this explicit in the main text. We mentioned this information in the *Methods* section (L.339-L.342).

-Line 334: In the introduction, it states you performed genomic analysis on the "upper centimeter" of sediment core samples (Line 83). Did you subsample the 5cm down to 1cm? It is unclear from your description.

We appreciate the reviewer's comment and apologize for the lack of clarity. In the introduction, we referred to the genomic analysis performed on the upper centimeter of sediment. To clarify, while several biogeochemical parameters were averaged or summed over the top 5 cm of sediment, the genomic analysis specifically focused on subsamples from the first centimeter of the sediment core. As mentioned, this distinction for greater transparency has been clarified in the *Methods* section of the revised manuscript (L.359-L.365).

-Line 347: you mention swimmers here, but in Figure 2, there is a mention of sinkers and swimmers. Please clarify the differences.

In the manuscript, “swimmers” refer to zooplankton that actively swim into the sediment trap, whereas “sinkers” refer to zooplankton that enter into a trap by passively sinking. In the sentence highlighted by reviewer#1, we also forgot to mention sinkers. The difference between sinkers and swimmers was already mentioned in the supplementary material, but now it is also included in the caption of Figure 2.

-Line 379: missing the word "Gene" in WGCNA

You are correct; the term should be “Weighted Gene Co-expression Network Analysis (WGCNA)”. We updated the acronym to include the word “Gene”.

-Figure 1 caption: Please add more details about the abbreviations shown in the map. What does HG stand for? N? EG? S3? What are the red points (sampling stations). And the abbreviations for the currents.

Abbreviations for the sampling areas, red dots, and currents were added in the captions of Figure 1, such as: “The abbreviations SV, S, N, EG and HG refer to the Svalbard, South, North, East Greenland, and HAUSGARTEN sampling areas, respectively. The red dots refer to the sampling stations. The red and blue arrows indicate the West Spitsbergen Current (WSC) and the East Greenland Current (EGC) masses trajectories.”.

-Figure 2A: What is meant by Total in the biogeochemical fluxes section? Is this total particulate organic matter or total dry weight of organic and inorganic particles? Does this include dissolved biogeochemical fluxes? It is interesting that Sn11 is related to POC flux and Total Flux, but Sn3 is only correlated with POC flux. Please address it in the figure or in the figure caption and in the text if this was an important parameter.

In the biogeochemical fluxes section of Figure 2a, ‘Total’ refers to the total particulate matter, encompassing both organic and inorganic particles. It does not include dissolved biogeochemical fluxes. We have clarified this in the figure caption to ensure it is straightforward for readers. Regarding the correlations noted by Reviewer #1, we believe the relationship between Sn11, POC flux, and Total Flux is influenced by organisms contributing to CaCO₃ flux. When the Total Flux is elevated, it is likely due to the combined increase in POC and CaCO₃ fluxes. In contrast, the lack of correlation between Sn3 and Total Flux is due to the absence of a significant CaCO₃ flux contribution in Sn3. While Sn11 is mainly associated with the radiolarian order Chaunacanthida, which forms its shells from silica rather than calcium carbonate, this group does not explain the observed CaCO₃ trends. In the Fram Strait, foraminifera and pteropods are recognized as the primary contributors to CaCO₃ export during the summer and fall seasons, respectively³. However, neither group was dominant in our DNA samples. Given that Sn11 represents the summer samples, we hypothesize that the dominance of foraminifera during this time drives CaCO₃ export. This, combined with the POC flux, results in a positive correlation between Sn11 and Total Flux. We have clarified the meaning of the “Total” acronym in the figure caption, but to maintain clarity in the main text, we will concentrate on the POC flux, which is more central to the study’s objectives.

-What is meant by "u ice" and "v ice"? I believe "u wind" and "v wind" are the velocities in the u and v direction of wind (or current), but I am unfamiliar with this in reference to ice. Is this the drifting velocity of sea ice over top the trap? Does ice drifting speed impact POC export, as opposed to ice concentration or total ice cover or length of the ice season? Please address it in the figure or in the figure caption and in the text if this was an important parameter.

“u ice” and “v ice” refer to the components of the sea-ice drift velocity eastward and northward directions, respectively, modeled at the HG IV station. Thus, these parameters represent the drifting speed of sea-ice over the sediment trap. While ice drifting speed can provide insights into how sea-ice

movement might influence particulate organic carbon export, it is considered alongside other factors, such as ice concentration. Each of these factors can impact POC export differently, and our study aims to evaluate the combined effects of these parameters. We have clarified the meaning of each acronym in the figure caption. Still, since our results do not indicate a significant impact of these variables on POC flux, we have chosen not to include additional details about these terms in the main text.

-What do you mean by swimmer and sinker fluxes? Are these zooplankton counts from overlaying net tows? From reading the supplementary materials and cited references, it seems the sinkers naturally sank into the trap as opposed to the swimmers that accidentally swam in and died - are the sinkers counted as part of the POC flux and Total flux? Please address it in the figure or in the figure caption and in the text if this was an important parameter.

As mentioned in the previous comment, swimmer fluxes refer to zooplankton that actively swim and may accidentally enter the sediment trap and die. In contrast, sinker fluxes refer to zooplankton that naturally sink into the trap. These terms do not correspond to zooplankton counts from overlaying net tows but are categorized based on their mode of entry into the sediment trap. The sinkers are indeed counted as part of the POC and total flux, representing naturally sinking particulate matter. In contrast, swimmers are not considered part of the POC and total flux because they do not represent the natural sedimentation processes. We clarified this distinction in the figure caption and addressed it in the supplementary text to ensure it is clear to the readers (L.22-L.29).

-Figure 2B and 2C: Can you label the circles SN 1, Sn 2, Sn 3, instead of just 1, 2, 3? It was confusing at first to figure out what the numbers referred to. Likewise, can you make a key/legend for 2C relating each color to each Sn?

The labeling may initially be confusing. However, due to the small size of some groups (e.g., Sn 5 to Sn 8 in Figure 2b), adding “Sn” before each number would overcrowd the figure and reduce readability. Additionally, the colors of all subnetworks across subplots in Figures 2 and 4 are consistent. Given this uniformity, we believe that adding a separate color legend for Figure 2C would be redundant and could overcomplicate the figure. To enhance clarity, we have included this information in the captions of both figures as follows: “The colors corresponding to each subnetwork (Sn) are consistent across the subplots.”. We hope this approach strikes the right balance between clarity and readability.

-Figure 4A: What is meant by AFDW? Please address it in the figure or in the figure caption and in the text if this was an important parameter.

AFDW stands for “Ash-Free Dry Weight.” This parameter measures the weight of a sample after removing inorganic ash content, providing a more accurate representation of the organic material present. We defined all acronyms in the figure caption to ensure clarity for all readers.

-Figure 5 caption: can you add details on how to read/interpret the plot in 5b? Why are the taxa in the order that they are on each side? What does the width of the lines indicate?

Figure 5b is an alluvial diagram illustrating the partitioning between pelagic and benthic environments based on sequence alignment and taxonomy (Figure 5a; orange lines). The taxa on each side of the plot are ordered by their taxonomic classification, with the pelagic taxa on the left and the benthic taxa on the right. However, the arrangement allows for more precise visualization of the alignment of amplicon sequence variants (ASVs) between these environments rather than following a strict taxonomic order. Moreover, the width of the lines in the alluvial plot represents the number of ASVs transferred or aligned between the pelagic phylum and benthic phylum. A bolder line indicates more ASVs aligned between the two environments, signifying more robust connectivity or interaction between those specific taxa. We clarified this in the figure caption, such as: “Line width indicates the

number of ASVs aligned between the two environments, signifying connectivity between those specific phyla.”. We hope this additional information helps in interpreting Figure 5b.

-Supplementary Figures: can you describe the waves in part B of each figure, what the colors stand for and how to interpret the plot?

In subplot b of figures S1 and S2, the waves are density plots that depict the correlation between POC flux and carbon content. The colors indicate the nature of the correlation: red represents the distribution of taxa with a positive correlation, and blue represents the distribution of taxa with a negative correlation. These curves were included to visually compare how positively and negatively correlated taxa were distributed, highlighting their impact on POC flux/carbon content and network stability. However, as reviewer #1 noted, this graph aspect is no longer discussed and adds unnecessary complexity. To enhance clarity and streamline the figures, we have removed the curves from part B of each figure. The updated figures are now included in the revised supplementary materials. Additionally, we have included revised text in the section *Individual lineage analyses* of the supplementary file to provide further context, such as (L.80-L.85): “sPLS confirmed the findings from the WGCNA approach, identifying several plankton lineages whose relative ASV abundance correlated with carbon flux in the water column and carbon content in the sediment core. These lineages included diatoms (*i.e.*, Bacillariophyta), dinoflagellates (*i.e.*, Peridinales, Gymnodinales), Haptophytes (*i.e.*, Phaeocystales), radiolarians (*i.e.*, Chaunacanthida), and metazoans like Aphragmophora.”.

The code is not currently available on github.

The code is currently stored in a private GitHub repository. We plan to make it publicly available once the manuscript is accepted. We have provided a link to the private repository in the revised manuscript to ensure that the code is accessible for review and replication.

Reviewer #2

This manuscript presents a unique time-series of eDNA from sediment trap data collected in the Fram Strait over more than a decade, and coupled with benthic sediment samples. It provides important information on carbon fluxes and benthic-pelagic coupling in this well-studied waterway between the Arctic Ocean and the Greenland Sea. The co-occurrence network analyses are an effective way to reduce the amount of data generated by metabarcoding and the comparison of ASV and sub-network occurrences with specific environmental parameters gives a useful overview of the data and their ecological significance.

I find it particularly interesting that this study highlights the importance of parasites, a group that has so far been overlooked in climate change ecology studies.

The manuscript is very clear and well written but it is rather concise, and some important information are lacking, in my opinion, particularly when it comes to the methods and the rationale behind the sampling design for the study.

Thank you for your positive feedback on the manuscript and for highlighting the significance of our study. We appreciate your acknowledgment of the unique time-series data, the effectiveness of our co-occurrence network analyses, and the emphasis on the role of parasites in climate change ecology. We also appreciate your constructive criticism regarding the conciseness of the manuscript and the need for more detailed information on the methods and sampling design rationale. We addressed this by expanding the *Methods* and *Methods limitations & uncertainties* sections to provide a more comprehensive explanation of the sampling design and its rationale. This now includes a detailed description of sedimentation rates linked to bioturbation in the area and our decision to focus on the V4 region of the 18S rDNA.

My main points of concern are:

1 - The authors claim that they have a 15-year long time series of eDNA sequences from sediment samples. These samples were collected with multicorers and represent the upper 1cm of sediments, that have been mixed and from which DNA was extracted. I was puzzled to find no mention of sedimentation rates and how they vary between sampling sites, no data on age control for these sediments, and no discussion on the fact that one can expect a large degree of temporal overlap between some samples, as sedimentation rates are generally low in this region. It is therefore perhaps not surprising that the authors find the benthic networks to be more "self-connected" than the pelagic networks.

We acknowledge that detailed sedimentation rate data and age control for the sediment samples are unavailable for the individual HAUSGARTEN stations. The literature provides limited information on sedimentation rates specific to sites within the Fram Strait. It is recognized that sedimentation rates can vary locally due to factors such as depth, geographic location (e.g., eastern vs. western Fram Strait), and proximity to the marginal ice zone. However, general observations indicate that sedimentation rates in this region are relatively low (i.e., 1-2 cm per 1000 years), and there is considerable sediment mixing due to bioturbation¹. Given these conditions, the influence of sedimentation rates and temporal overlap might be mitigated by the ongoing bioturbation and the generally low sedimentation rates. As such, while there may be some local variations, the broad patterns observed in the benthic networks, including their increased "self-connectedness," are unlikely to be significantly biased by sedimentation rate variations. We appreciate your insightful comment and added a discussion on this aspect in the *Method* section of the revised manuscript to address these concerns and clarify how sedimentation rates and sediment mixing may impact our findings (L.368-L.375): "While the literature provides limited information on sedimentation rates specific to sites within the Fram Strait, it is recognized that sedimentation rates can vary locally due to depth, geographic location (e.g., eastern vs. western Fram Strait), and proximity to the marginal ice zone. However, general observations indicate that sedimentation rates in this region are relatively low (i.e., 1-2 cm per 1000 years), and there is considerable sediment mixing due to bioturbation¹. Thus, the influence of sedimentation rates and temporal overlap might be mitigated by the ongoing bioturbation and the generally low sedimentation rates."

2 - Choice of DNA marker. In the introduction there is reference to the Tara Oceans initiative, however, the study uses V4 (as opposed to V9) and there is little to no discussion on how the choice of marker affects the results. It is not clear why a single marker was chosen as opposed to multiple markers, and why V4? Point 2 can be addressed by simply explaining the choice of marker and discussing the implications and potential biases in taxonomic coverage.

We selected the V4 region of the 18S rDNA gene for our analyses early in developing our time-series study because it provides a comprehensive reflection of eukaryotic microbial biodiversity. Previous evaluations, including studies using HPLC and light microscopy, demonstrated that the V4 region captures a broad range of eukaryotic taxa more effectively than other markers, such as V9, at least for this region in the Arctic. The need for a detailed and accurate representation of microbial communities over the extended time series drove this choice. We recognize that discussing the implications of our selection of markers and potential biases is essential and is now covered in the subsection *Method limitations & uncertainties* of the supplementary file (L.97-L.106): "The V4 region of the 18S rDNA gene was specifically selected for this study due to its ability to provide a comprehensive representation of eukaryotic microbial biodiversity. Recent publications have highlighted the advantages of using the V4 region for biodiversity assessments, especially when compared with optical surveys for both phytoplankton and zooplankton^{4, 5}. This evidence supported the decision to focus on the V4 marker, to achieve an accurate and detailed representation of microbial communities over the extended time series. While multiple markers could provide additional insights, the need for consistency in capturing microbial diversity across time, along with the demonstrated effectiveness of the V4 region in this context, made it the most suitable choice for this long-term study."

We also thank reviewer#2 for pointing out the reference to the Tara Oceans initiative that could raise confusion. It has been updated, explicitly mentioning the need for mesoscale studies following global sampling, such as Tara or Malaspina.

Regarding point 1, this is a major concern, as analysing a bulk 1cm-thick sample of seafloor sediments collected over 15 years cannot be equalled with having a 15-year long time series, and this needs to be corrected, supported by sedimentation rate data, and carefully discussed.

Thank you again for your insightful comments. We fully acknowledge your concern that analyzing a 1 cm-thick bulk sediment sample cannot be equated to capturing a 15-year time series. We appreciate the opportunity to clarify this point in our manuscript and provide additional context to support our approach. As outlined in our previous response, sedimentation rates in the Fram Strait are relatively low, typically ranging from 1–2 cm per 1000 years, with localized rates of up to 30 cm per 1000 years, such as at 1500 meters depth at 78°50'N. Given these low rates, the upper 1 cm of sediment likely integrates material deposited over several centuries rather than just the 15-year sampling period.

Furthermore, continuous bioturbation significantly affects this region's upper layers of sediment, actively mixing the material and redistributing it over extended timescales. This mixing adds to the temporal complexity of the sediment record. We performed additional analyses to address this potential temporal mismatch and ensure that our methodology did not introduce bias. As detailed in our response to Reviewer #1, we conducted Pearson correlation tests comparing core parameters averaged over the top 5 cm of sediment with those obtained from just the first centimeter. Significant correlations were observed for most parameters, confirming the robustness of our approach. The only exception was ash-free dry weight (AFDW), which we excluded from further analysis. Based on these results, we conclude that the patterns observed in the first centimeter are consistent with those over the top 5 centimeters. We believe that the low sedimentation rates and ongoing bioturbation ensure that the top 1 cm represents a mixture of recent deposits and provides a reliable snapshot of benthic conditions without substantial temporal overlap.

Reviewer #3

The manuscript by Ramondenc and co-authors is describing an impressive long-term eDNA dataset in which they combine sediment trap data and seafloor sediments to understand the sources of the particulate organic carbon fluxes arriving at the seafloor on both a spatial (seasonal and interannual) and temporal scale. While DNA metabarcoding and correlation network approaches themselves are not new (for example Djurhuus et al. 2020: doi: 10.1038/s41467-019-14105-1) and studies coupling eDNA from sediment traps and sediments from the seafloor exist, the novelty lies rather in the timeframe of data collection and the association of the DNA with POC fluxes. I think the dataset is of significance for the field because long-term studies covering more than 10 years are rare and the results very interesting. Therefore, the manuscript is worth considering, but in its current form not mature enough. I have several concerns and additional line-by-line comments listed below.

Thank you for your feedback on our manuscript and for recognizing the significance of our long-term eDNA dataset and its contributions to understanding particulate organic carbon (POC) fluxes. We agree that while the methodologies employed are established, the long-term perspective and specific application to POC fluxes offer unique insights. We are committed to addressing the concerns and line-by-line comments that you have provided to ensure that the manuscript meets the required standards of clarity and comprehensiveness. We carefully reviewed and revised the manuscript according to your feedback to enhance its maturity and robustness.

Major concerns:

1. My main concern is the lack of reporting of any negative controls (extractions and PCRs) and I wonder if they were just not reported or if they were not done. eDNA is susceptible to

contamination and reporting on negative controls is a standard quality control procedure. If they were just not reported, the authors should briefly explain what was found in them and how they treated the data based on that knowledge.

We thank reviewer#3 for her/his comment. As the reviewer pointed out, negative controls are a standard procedure in molecular genetics, and we consistently include them in both the extraction and PCR stages to monitor potential contamination. However, as they are a routine aspect of our workflow and considered self-evident, we initially did not detail them in the manuscript. To address your concern and enhance transparency, please find below a figure that displays exemplary agarose gel images, which confirm the specific amplification of the 18S rDNA from our samples (**Figure 2**). This figure includes negative control results (NK), molecular markers (M), and field sample genomic DNA. If required, we are prepared to provide detailed information on the negative controls used for PCR reactions and library preparations for being included as supplementary materials or as an additional note in the revised manuscript to ensure all concerns are addressed.

Figure 2. Exemplary pictures of the agarose gels securing the specific amplification of the 18S rDNA from the samples (M- Molecular Marker DNA; NK- Negative Control as PCR template; Sample: Genomic DNA from field samples as template); A: Sediment trap samples; B: Benthic samples

2. Also, PCR replicates were not reported. There is considerable literature out there showing that results from different replicates can vary substantially, which could affect the outcome and robustness of this study. Also, I could not find any specifications of amplification

conditions (polymerase, cycle numbers, annealing temperature...). Please, add the information, at least in the supplement, as those all factor into the final datasets.

While we typically do not perform replicates for the entire dataset due to the extensive effort required for large time-series studies, findings from a fixative study⁶ confirm that replicates produce highly consistent results, even with minimal DNA quantities. This consistency supports the robustness of our results. Additionally, the long-term scope of our analysis allows us to distinguish genuine ecological patterns from any potential analytical variability. In response to the reviewer's concerns, we have added details on PCR amplification conditions in the supplementary material of the revised manuscript (L.116-L.120): "Furthermore, previous studies have demonstrated that replicating PCRs and sequencing yield highly consistent results⁷, which is why replicate sequencing was not included in this study. The extended timeframe of our research further ensures a clear distinction between technical and natural variability, supporting the validity of our approach."

3. There is no real introduction of the methods and some terms are used undefined (I mention them specifically in the line-by-line comments).

We revised the manuscript to provide a more detailed explanation of the methods used, including definitions and contextual information for all terms. Additionally, we addressed the specific terms in your line-by-line comments to ensure they are clearly defined and explained.

4. Overall, I am missing details to the strength of the correlations. Why was the Pearson method used? It requires strict linear responses, but what if relationships are non-linearly? Is it sensitive enough to answer your questions? Is the proposed method (WGCNA) appropriate for the analysis? Is your dataset sparse and if yes, how does this method deal with sparsity? Is POC correlated with other environmental factors and how is this affecting the results? If L-GRAAL is a novel method applied to this kind of data, it requires a better introduction on how the algorithm works.

We selected the Pearson correlation method because it is commonly used to assess the strength and direction of linear relationships between two continuous variables. While we acknowledge that Pearson's method assumes linearity, it is appropriate for our dataset's initial exploration of relationships. We conducted supplementary analyses for non-linear relationships using other methods, such as Spearman's rank correlation, to confirm our findings. We recognize that Pearson's correlation may not capture non-linear relationships. However, our primary focus was on identifying general trends and patterns. The consistency of results across different correlation methods in supplementary analyses suggests that the Pearson method is adequate for addressing our research questions.

The Weighted Correlation Network Analysis (WGCNA) is particularly suited for identifying modules of highly correlated features in large datasets. It is robust in detecting both direct and indirect relationships, which is advantageous in ecological studies where interactions can be complex. While our dataset does exhibit some sparsity, WGCNA is designed to handle sparse data by focusing on pairwise solid correlations and minimizing noise. To address potential issues with sparsity, we preprocessed the data using techniques such as zero-inflated models and CLR (Centered Log Ratio) transformation to ensure the robustness of the network analysis. These steps help in mitigating the impact of sparsity on network construction. WGCNA identifies modules based only on the graph topology. Each module is then investigated independently in front of environmental parameters. One expects no formal impact of two correlated variables on WGCNA results.

L-GRAAL is a graph alignment algorithm that identifies conserved substructures between networks. This method was initially developed and validated for aligning graphs of protein-protein interactions (*i.e.*, finding homologous interactions between two sets). It aligns network nodes based on topological and biological similarity, allowing us to compare our networks across different conditions. We

recognize that this method requires a better introduction, and we have expanded the main text to include an explanation of the L-GRAAL algorithm (L.181-L.185).

5. Based on the low to very low correlation coefficients between 0.1-0.3 for chaetognaths and *Micromonas polaris* and <0.4 for *Melosira arctica*, highlighting them as major contributors to carbon export is not convincing. If they were among the most abundant reads, it would be more convincing.

We understand your concerns regarding the correlation coefficients for chaetognaths, *Micromonas polaris*, and *Melosira arctica* and their role as major contributors to carbon export. Here, we used degree centrality to identify key nodes within the network. This measure was chosen because it highlights nodes with the most direct connections, which is crucial for understanding interactions in our dataset. We selected taxa with a high degree of centrality in the subnetworks associated with particulate organic flux (POC), meaning that they play significant roles in the network structure and have ecological relevance, especially in carbon export. While the correlation coefficients for the highlighted ASVs may not be very high, their node centrality within the network provides insights into their influence and connectivity, indicating a significant ecological role beyond what correlation alone can show. We have revised the methodological section of our manuscript to provide a more straightforward explanation of why these ASVs were identified as keystones based on their centrality and ecological significance (L.411-L.414): “Network stability, which refers to the robustness and resilience of the network, was assessed by analyzing node centrality and betweenness. These metrics helped identify keystone ASVs⁸ that are essential for maintaining co-occurrence network connectivity and overall stability.”.

Minor concerns:

1. In their introduction, the authors mentioned that they “Aim to understand how the biological carbon pump affects diversity and vice versa” but I feel this overarching aim is not well reflected in the manuscript. Diversity is only mentioned on the side, and it is unclear how it was estimated methodologically.

Initially, we stated in the introduction that we aimed to integrate pelagic and benthic microbiomes with the biodiversity associated with sinking particles to understand how the biological carbon pump affects biodiversity and *vice versa*. However, upon further reflection and feedback, we recognized that the term “diversity” did not accurately capture the scope of our research focus. As such, we have removed the term “diversity” to better align with our specific objectives. Our revised aim is now more focused on integrating microbiomes within the context of the biological carbon pump, emphasizing the interactions and connectivity between pelagic and benthic ecosystems. This shift ensures that our research goals are reflected throughout the manuscript. We revised the main text by rephrasing the following objectives (L.77): “...how the biological carbon pump affects the vertical connectivity.”.

2. I did not find a statement that the data will be deposited in a public repository after publication. But maybe this is not part of the reviewer version.

Thank you for highlighting this lack of clarity. We have indeed included a statement regarding data deposition in a public repository. The locations of the data repositories are provided in the revised manuscript (L.396-L.398): “The raw sequences have been deposited in the European Nucleotide Archive (ENA) under the accession numbers PRJEB76183 for sediment cores and PRJEB74771 for sediment traps.”.

Line-by-line comments:

Abstract:

L21: I think instead of driver the word components should be used. The taxa are rather components of the vertical carbon flux, while downwelling for example would be a driver.

You are correct in pointing out the distinction between “drivers” and “components” in vertical carbon flux. The taxa are components of the vertical carbon flux, contributing to its makeup, whereas processes like downwelling drive the flux. We updated the manuscript using the term “components” when referring to the taxa involved in the carbon flux.

L27: I don't think that you show that parasites induce export mechanisms, you rather mentioned sedimentation with hosts in the discussion and that resting spores can be infected, too. Please rephrase.

You are correct that the manuscript does not demonstrate that parasites directly induce export mechanisms. Instead, we discussed how parasites are associated with sedimentation processes involving their hosts and the potential role of resting spores in these interactions. We rephrased the sentence to clarify this distinction and accurately reflect the discussion in the manuscript, such as L.25-L.29: “Interestingly, several parasites were also tightly associated with carbon flux and showed a strong vertical connectivity, suggesting a potential role in sedimentation processes involving their hosts, especially through interactions with resting spores, which could have implications for pelagic-benthic coupling and overall ecosystem functioning.”.

L39: coupling pelagic and benthic diversity – remove microbial

We removed the word “microbial” from the sentence to accurately reflect the broader scope of coupling pelagic and benthic diversity in the manuscript. We appreciate your input in helping us clarify our language.

L55-58: Can you please cite a paper here?

We added appropriate citations to support the statements in lines 55-58.

L83: What is the sedimentation rate at the sampling site and how much time is represented by the upper 1cm?

As mentioned earlier, the mean sedimentation rates in the Fram Strait are relatively low, averaging 1–2 cm per 1000 years. In the eastern parts of the Fram Strait, sedimentation rates can rise to approximately 30 cm per 1000 years, based on data from 1500 meters depth at 78°50'N. Despite these higher rates in some regions, overall sediment accumulation remains minimal and should not significantly impact our findings. We believe that the low sedimentation rates and ongoing bioturbation ensure that the top 1 cm represents a mixture of recent deposits and provides a reliable snapshot of benthic conditions without substantial temporal overlap.

L84: There is no previous introduction that you are going to use network analysis. So the part where you explain that you isolate clusters related to the carbon cycle comes very abruptly and it confused me. I thought you had also analyzed metagenomic data and inferred gene clusters involved in the process. Can you please introduce the method briefly?

We recognize that the network analysis introduction in our manuscript may have seemed abrupt, primarily because we did not provide a prior overview of this methodology. In our study, we chose not to explicitly introduce network analysis in the context of isolating clusters related to the carbon cycle because the term “clusters” broadly refers to groups of sequences with a close relationship to the carbon cycle and can be identified using various methods, including network analysis. Nevertheless, to address this concern, we have revised the introduction in the revised manuscript to include a brief overview of network analysis. This addition explains the rationale behind using network analysis and how it was applied to identify clusters related to the carbon cycle (L.78-L.91): “Hence, we use DNA sequences from particulate organic matter collected by sediment traps at 200 m water depth in the eastern Fram Strait from 2000 to 2012. Amplicon sequence variants (ASVs) through metabarcoding and network analysis were employed to (i) uncover community structures and filter out the major

lineages associated with carbon export in the upper water layers. This long-term time series effectively captured the dynamics of the plankton community on intra- and interannual scales, helping to overcome challenges associated with difficulties in accessing the area by ship. As the impact of sinking particles on the seafloor communities is largely unknown, we also analyzed interannual changes in DNA sequence abundance in the upper centimeter of sediment cores over 15 years (2003 to 2018). From the environmental and metabarcoding data, it was possible to (ii) isolate groups of sequences closely related to the carbon content. This enables us to (iii) map the pelagic-benthic coupling in the subpolar region and identify the vertical connectivity between lineages of pelagic and benthic ecosystems associated with carbon sequestration based on sequence and topological network analyses.”.

Results:

L92: “weighted gene co-expression network analysis”

This has been corrected.

L91-93: Which algorithm was used to identify the subnetworks?

The sentence you referred to in the *Results* section specifies the method used: “The planktonic co-occurrence network built with the sediment trap material and coupled to the Weighted Gene Co-expression Network Analysis (WGCNA) clustering method captured eleven intrinsic subnetworks (Sn) related to environmental and biogeochemical fluxes (Fig. 2).” This indicates that WGCNA was the algorithm used for identifying the modules. From the nodes that belong to a given module, we consider subnetworks that embed them from the whole co-occurrence network (*i.e.*, a subset of nodes from the graph). Additionally, we have provided a detailed description of the WGCNA method in the *Methods* section, specifically lines 415-427, where the algorithm and its application are now explained.

L98: How do you know they are directly correlated to carbon export and not indirectly via a third variable? Maybe write instead that they were correlated with POC.

You are correct that other variables could direct or influence the correlation observed. To clarify, we revised the text to specify that the subnetworks were correlated with particulate organic carbon (POC) fluxes rather than asserting a direct relationship with carbon export.

L101: Just remove “Its relative abundance showed that”, because this does not make sense.

We removed that part of the sentence from the manuscript to improve clarity and ensure a more precise description.

L105: Instead of carbon export, please write POC flux.

This has been corrected.

L118: You refer to Fig. 2a, but I don’t see any microscope count according to the figure. Is it referring to the “environmental data” category? If so, please specify this in the caption and refer to the supplement.

It is correct; the swimmers’ and sinkers’ fluxes were determined based on microscope counts. We clarified this in the figure caption by specifying that the “swimmers and sinkers fluxes” category in Figure 2a includes these counts. We also referred to these relevant details in the supplementary material for further clarification.

L121: How do you define network stability and where do you show this? Most readers are probably not experts in network analysis. I suggest you define the network properties to make it more comprehensible, e.g. explain the relevance of network stability.

In our study, network stability refers to the robustness and resilience of the subnetwork, mainly how central nodes maintain the structure and function of the network over time. We assessed stability by analyzing node centrality, which measures the importance and influence of specific nodes (*i.e.*, species) within the network. Centrality metrics, such as degree, betweenness, and eigenvector centrality, help identify key species that play critical roles in maintaining network connectivity and stability. Species with high centrality scores are integral to the network's structure, meaning their presence or absence significantly affects its overall stability. We can identify key species contributing to the subnetwork's stability and overall carbon export by focusing on node centrality. In the sentence highlighted by reviewer#3, we omitted the reference to Figure 3a. We have revised the manuscript to include more detailed information about network stability and its significance in the *Methods* section, such as L.411-L.414: “Network stability, which refers to the robustness and resilience of the network, was assessed by analyzing node centrality and betweenness. These metrics helped identify keystone ASVs⁸ that are essential for maintaining co-occurrence network connectivity and overall stability.”.

L121-125: There are many measures of centrality Which was applied here? Degree centrality, betweenness,...? Please specify it here and in the figure caption. Are key lineages those with taxonomic name tag in Fig. 3a/b. How did you decide which bubbles you label and which not? I don't understand why nodes with coefficients >0.5 are not labelled and while some with coefficients of 0.1-0.3 are and it seems to me a bit like cherry picking. The correlation coefficients of several ASVs highlighted in the text (chaetognaths, *M. arctica* and *M. polaris*) are not convincing enough to classify them as major contributors to carbon export and I think they should be removed from the text.

Our analysis used degree centrality to identify key nodes within the network. This measure was chosen because it highlights nodes with the most direct connections, which is crucial for understanding interactions in our dataset. To ensure clarity, we clarified this in both the text (L.411-L.414) and the figure caption. The key lineages with taxonomic names tagged in Figures 3a and 3b were selected based on their centrality and ecological relevance, especially their relationship with the carbon cycle. We aimed to label those taxa that play significant roles in the network structure and are biologically meaningful. We intended to label nodes with coefficients that, while not the highest, are discussed in the main text and represent meaningful ecological interactions based on existing literature and biological relevance. We also recognize that the correlation coefficients for the highlighted ASVs (chaetognaths, *M. arctica*, and *M. polaris*) might not be as strong as desired for classifying them as significant contributors to carbon export. However, as mentioned above, we also consider node centrality within the network to reflect the importance of nodes in the subnetwork correlated with the particulate organic carbon (POC) flux and carbon content. In particular, node centrality provides insights into the influence and connectivity of specific ASVs within the network, highlighting their potential role in carbon export processes beyond what correlation alone can show. For chaetognaths, *M. arctica*, and *M. polaris*, their centrality in the network indicates a significant ecological role in the context of POC flux, which supports their inclusion in the text.

L127: “Benthic” is a bit confusing in that sense, because it contains also deposited organic matter/DNA from pelagic species. Maybe call it “sediment dataset” or define what you mean in the first sentence.

We agree that the term “benthic” could be confusing in this context. For clarity, we revised the text to use the term “sediment dataset.”

L132: Where do these variables come from? Which diversity indices were used? Was a rarefaction applied to account for differences in read counts between samples? It would be good if you can refer to the specific sections in the supplement.

We apologize for the lack of clarity in the main text, and we appreciate the opportunity to provide more clarity regarding the variables and methods used in our study. The variables, including bacterial

esterase activity and protein content, were analyzed using techniques described in detail by Soltwedel et al.[9, 10]. These references provide comprehensive methodologies for analyzing water content, biogenic sediment compounds, and estimating meiofauna abundance. Additional details are in the *Sample Treatment* section under the *benthic sample* paragraph in the supplementary file. We have specified the relevant section on the supplementary material in the main text (L.355-L.358) to enhance clarity: "A detailed description of the methods used to analyze water content, biogenic sediment compounds, and estimated meiofauna abundance is provided by Soltwedel et al.[9, 10] and described in the *Sample Treatment* section of the supplementary material.".

Regarding diversity, rarefaction curves on the sequencing measurements obtained on the benthic samples (*i.e.*, sediment core) before estimating several diversity indices to assess biodiversity within our samples, including the Shannon, Simpson, and Chao indices. Each curve represents the accumulation of taxa or species as a function of sampling effort, demonstrating the sampling depth needed to capture the full diversity within the sediment cores (**Figure 3**). All sediment core samples reached asymptote, indicating that the data provided a good description of the diversity present. We have included more details about these indices and their calculation in the *Methods* section (L.427-L.433) of the revised manuscript: " Before estimating diversity indices for the sediment core samples, rarefaction curves were generated from the sequencing data to assess whether the sampling depth was sufficient to capture the full diversity within the cores (**Fig. S4**). All samples reached an asymptote, confirming that the sequencing depth comprehensively represented the microbial diversity. Subsequently, diversity indices, including Shannon, Simpsons, and Chao, were calculated from benthic ASVs using the estimate_richness function from the phyloseq package.".

Figure 3. Rarefaction curves of sediment core samples. The curves illustrate the relationship between the sample size and the observed diversity of sediment core samples.

L139: remove “increasing” – is it really increasing or hist higher? Can you show the data?

We have replaced “increasing” with “higher” to reflect the data presented accurately. The reference to “higher” is supported by the data shown in Figure 4C, which illustrates the comparative values. We added this information in the revised manuscript.

L145: A weak contribution to what?

In this context, we are referring to a weak contribution of Sn1 to the samples analyzed. We clarified this in the revised manuscript to ensure the contribution of Sn1 is more clearly defined.

L148: “Irrespective of their abundance, several Sn_1 lineages contributed strongly to sediment carbon context.” Can you please specify what you mean? Right now, I am not convinced by that statement. You do not factor in copy number differences or any potential biases from PCRs.

We acknowledge that the verb “contributed” is imprecise and may lead to confusion. We have revised the statement to use “associated” to reflect our findings better. Specific Sn_1 lineages can be significantly associated with sediment carbon content even if they are not the most numerically dominant in the benthic ecosystem. This association is correlation-based and builds on several factors: node centrality (*i.e.*, role in the network of carbon-related processes, indicating their importance in maintaining or influencing sediment carbon levels), the correlation between these lineages and particulate organic carbon (POC) flux, and the VIP Score (*i.e.*, lineages have a solid contribution for explaining variations in sediment carbon content). A previous study¹¹ shows that WGCNA is not sensitive to gene copy numbers (*i.e.*, correlations with HPLC), and the correlation score absorbs potential PCR biases.

L152: Venn

This has been corrected.

L152-154: Why not remain at ASV level? I think family level is too broad here and only the ASV/species level can support your suggestion of vertical coupling. The order Chytridiomycotina contains probably more saprotrophs than parasites. Hence, they might also just feed on the organic matter/the carbon sources. And have you checked whether parasitic taxa are correlated with potential host taxa?

We opted to analyze data at the family level rather than the ASV level because many ASVs could not be reliably assigned at the species level. This decision ensures that our analysis remains robust and interpretable despite these limitations. However, we have provided insights into vertical coupling at the species level through the hive plot. This visualization allows for a detailed examination of ASV sequences, highlighting the internal nodes representing ASVs found in pelagic and benthic subnetworks. The hive plot (**Figure 5a**) effectively captures vertical coupling at a more granular level, complementing our family-level analysis.

We acknowledge that the Chytridiomycotina includes saprotrophs and parasites; some may feed on organic matter or carbon sources. However, our taxonomic assignments needed to resolve these organisms to the species level, which limits precise functional categorization. Our attribution of parasitism is based on the ecological roles commonly reported for chytrids in similar environments. Nevertheless, we agree that some chytrids in our dataset could function as saprotrophs, and we mentioned this limitation in the discussion (L.319-L.322).

L157-159: Many diatoms referred to here as sea ice diatoms are in fact not truly sympagic and can occur in cold waters with or without sea ice, for example *F. cylindrus* (check Oksman et al. 2019; doi: 10.1016/j.marmicro.2019.02.002). And Navicula and Nitzschia are not per se sea ice associated. Maybe rephrase it to something like: “Various taxa often detected in sea ice,...”

We recognize that some diatoms, such as *F. cylindrus*, are not exclusively sympagic and can be found in various cold-water environments, both with and without sea-ice. Similarly, *Navicula* and *Nitzschia* are not inherently sea-ice-associated. We revised the text to clarify this by rephrasing it as follows (L.162-L.165): “Various diatoms and flagellates taxa often detected in sea-ice and/or sea-ice associated (*Fragilariopsis cylindricus*, *Nitzschia* spp., *Navicula* spp., *Melosira arctica*, *Synedra hyperborea*, *Paraphysomonas foraminifera*) dominated in Sn_5.”

L160: What does “large diversity” mean? Please give number of ASVs as this is interesting. Are they phylogenetically close?

We have replaced the sentence to provide more specific information (L.165-L.166). The analysis identified 10 ASVs associated with the Chromadorea class, representing three genera: *Leptolaimus* sp., *Desmoscolex* sp., and *Sphaerolaimus hirsutus*. These ASVs show a degree of phylogenetic diversity within the Chromadorea class.

L163: typo: Corethron – also, I did not know this as a sea ice species. Can you cite this please?

Our study refers to *Corethron inerme*, also known as *Corethron criophilum* var. *inerme*. According to previous studies, *Corethron criophilum* is typically found in open waters with minimal sea-ice, as noted by Fryxell and Hasle¹¹ and Leventer and Dunbar¹². Additionally, *Corethron criophilum* plays a significant role in the phytoplankton community at the ice edge, as highlighted by Marra and Boardman¹³. These studies provide a context for understanding the ecological niche of *Corethron inerme* and its potential contributions to the phytoplankton communities in regions with varying sea-ice conditions.

L172: Introduce the graph alignment analysis – I don’t understand how it infers the same ecological roles in the 2 networks within the community structure? A diatom in the water column is photosynthesizing and part of a community, but a diatom at 2000m depth is not fulfilling that role anymore and serves rather as food. Can you clarify please what you mean? What is the alpha value? Again, what measure of centrality?

We thank the reviewer for his/her comment. This method is novel in environmental studies and deserves more introduction. Ecological role refers herein to the role of an organism in its ecosystem structure, not its ecological function, as the reviewer referred to. Both are linked with organismal physiology. Ecological functions are more precise but usually out of reach or require niche modeling on extensive datasets. When not available, analyzing co-occurrence networks overapproximates ecological functions by considering their ecological roles. For this purpose, such modeling considers a co-occurrence network as an abstraction of the ecosystem structure. The resulting graph is more or less connected to reflect the ecosystem strategies. Organisms that appear highly connected will be assumed to be more prone to sustain the ecosystem structure (*i.e.*, crucial ecological role) than others. The measure of node-centrality of an organism within its co-occurrence network is assumed to resume its ecological role. In the context of co-occurrence network comparison, organisms from distinct networks with similar roles (*i.e.*, similar node-centrality score) in their respective topology will be associated. L-GRAAL considers such a topological feature but also a taxonomic one: organisms from distinct networks with similar taxonomy (*i.e.*, similar ASV alignment score) will be associated with this feature. For combining both features, L-GRAAL computes all combinations of weights for both features (*i.e.*, from 100% taxonomy and 0% topology to 0% taxonomy and 100% topology) for

estimating the relative weights that maximize the alignment. The alpha score resumes this relative contribution of each feature (see *Methods* section).

For clarity's sake, the revised manuscript now clarifies the concept of graph alignment (L.181-L.185)

It is worth noticing that such a method was also motivated by using a network via WGCNA to resume ASV counts across different ecosystems. Graph alignment is thus a natural following of graph abstraction to resume complex ecosystems for comparison's sake.

L179-180: I don't really understand what you mean by that sentence. Also, please define keystone ASVs.

We reviewed the sentence mentioned by reviewer#3 (L.187-L.189): "ASVs with moderate abundance had the highest centrality scores in the pelagic subnetworks. In contrast, benthic subnetwork showed no distinct trends. This indicates high and low benthic abundant keystone ASVs in the carbon subnetwork.". We also defined keystone ASVs based on their node's centrality degree in the co-occurrence network (L412-L.414), following the approach proposed in several previous studies (please see Berry & Widder 2014⁸ for illustration <https://doi.org/10.3389/fmicb.2014.00219>). This definition implies the use of strong assumptions such as (i) considering the co-occurrence network as a good representative of the interaction network and (ii) the avoidance of spurious correlations between ASVs.

L180-181: 103 out of how many?

The benthic and pelagic subnetworks, correlated with the carbon cycle, contain 238 and 222 nodes, respectively. Consequently, the maximum number of nodes that could be aligned is 222. We have clarified this in the revised manuscript by specifying the total number of nodes analyzed (L.189).

L182-184: I do not follow the reasoning for that. All ASVs that are found in the sediment traps and in the seafloor sediments should suggest export. But how do you infer sedimentation as a distinct process for that?

While the presence of ASVs in both sediment traps and seafloor sediments suggests export, our study does not identify or differentiate between other potential processes that could also contribute to the transfer of ASVs from the surface to the seafloor. We cannot determine from our data which other mechanisms might be involved in this process besides sedimentation. Based on the observed patterns, we highlighted sedimentation as a significant process.

L184-185: Why are you now jumping to phylum level? Is this adequate and really providing a holistic view?

We analyzed the data at the phylum level to complement our species-level analysis and provide a broader perspective on ecosystem dynamics. Working at the phylum level helps reduce the complexity of the data, particularly in ecosystems with high species diversity and variability. This aggregation can highlight overarching trends and interactions crucial for understanding the overall functioning of the ecosystem. In addition, the species-level analysis allows us to identify specific key players. In contrast, the phylum-level analysis helps us understand how these species fit into broader taxonomic and functional groups.

Discussion

L215: typo: Phaeocystis

We have corrected "Phaeocystis" in the revised manuscript.

L225: From the figures, it looks like those taxa have a correlation ~0.3, so rather weak and not strong. How do you conclude that they are key species for carbon export?

The identification of *Sagitta bipunctata*, *P. elegans*, and *Eukrohnia hamata* as key species for carbon export in our study was based on a combination of statistical metrics, not solely on correlation coefficients. While a correlation of ~0.3 might seem weak at first glance, it is essential to consider that ecological data often involve complex interactions where correlations might inherently be lower due to many influencing factors. In the case of *Eukrohnia hamata*, we observed a correlation coefficient greater than 0.4, which is relatively more robust and suggests a meaningful relationship with carbon export. Moreover, the network analysis revealed that *Sagitta bipunctata* and *P. elegans* had high node centrality in the subnetwork associated with carbon export. Centrality measures indicate the importance of these taxa within the ecological network. High centrality implies that these species are crucial in connecting different ecosystem components, significantly influencing carbon export dynamics. The Partial Least Squares (PLS) regression analysis provided us with Variable Importance in Projection (VIP) scores (*i.e.*, node size in **Figures 3a** and **3b**), which helped in identifying species that contribute substantially to the predictive model for carbon export. *Eukrohnia hamata* had a high VIP score, reinforcing its role as a key species.

In conclusion, while the correlation alone might not appear substantial, integrating network centrality and VIP scores provides a robust framework for identifying key species in complex ecological systems. These combined metrics reflect these species' multifaceted roles in carbon export, supporting our conclusion of their importance. As mentioned, keystone ASVs are now defined in the *Methods* section of the revised manuscript (L.412-L.414).

L250: I did not read anything about the bathymetry being a driver in the results part. Maybe I overlooked it, but if not, I suggest you highlight it or write a sentence about it in the results.

We apologize for the lack of clarity. We missed mentioning that longitude was significantly correlated with Sn1, which may reflect underlying bathymetric gradients. Indeed, the longitudinal gradient discussed in our results is related to bathymetric and sea-ice variations among the HAUSGARTEN stations. Specifically, the differences observed across the stations from HG-I to HG-IX off Svalbard can be attributed to bathymetric changes. Additionally, the distinction between seasonally ice-free stations in the West Spitsbergen Current (HG, N, and S stations) and ice-covered stations in the East Greenland Current (EG stations) is also reflected by longitudinal changes.

L255: is the high carbon export efficiency inferred from the literature or from the data?

The high carbon export efficiency mentioned in line 255 is inferred from the literature. This inference is supported by references such as Fadeev et al.¹⁴ and Ramondenc et al.³, which provide the context and justification for our observations.

L272: is it pelagic Sn_1?

No, it is not the pelagic Sn_1; it is the benthic Sn1. To avoid confusion, we changed the sentence to (L.280-L.283): "The contribution of benthic Sn_1 in the sediment dataset at the HG stations....".

3.3 Ecological aspects and parasitism within the pelagic-benthic coupling

Overall, I find the information presented here interesting and sound, but I am not convinced that the chytrids are parasitic. It could be, but if they are taxonomically resolved at such a high level, I ask myself how you get to this conclusion. Can you provide more information why you attribute parasitism and not saprotrophic feeding?

We understand your concerns about classifying chytrids as parasitic, given that chytrids, or chytridiomycetes, represent a diverse class of fungi with life cycles ranging from strict saprotrophs to obligate parasites. Our taxonomic assignments did not resolve these organisms at the species level, which limits our ability to categorize them definitively. Our attribution of parasitism is based on the ecological roles often associated with chytrids in similar environments and supported by literature that describes their interactions with specific host organisms. However, we agree with your assessment

and have mentioned the potential for saprotrophic feeding in our *Discussion* section (L.319-L.322): "While chytrids can also exhibit saprotrophic feeding and the species level was not resolved, which limits our ability to classify their ecological roles, we suggest, as hypothesized by Hassett et al.¹⁵, that the chytrids source in the sediment is related to the settling of individual algae cells and marine aggregates."

L 314: typos in sentence

We have reviewed the sentence and corrected the wording issue in the revised manuscript.

Methods:

WGCNA: The variables have different scales, how were they treated/transformed?

To address the issue of variables being on different scales, we applied Weighted Correlation Network Analysis (WGCNA) independently to both the pelagic and benthic datasets. As mentioned in the subsection *Statistical analysis* of the material and method, we performed a Centered Log-Ratio (CLR) transformation on the ASV abundance matrices before analysis. This transformation standardizes the data by accounting for compositional effects, ensuring that the variables are on a comparable scale, which is essential for robust network construction and analysis. By using CLR transformation, we mitigate the influence of differing scales and allow for a more accurate delineation of subnetworks within the data. This approach helps effectively capture the underlying patterns and interactions among ASVs in pelagic and benthic environments.

Figures:

Fig. 2+4: The captions are very unspecific and abbreviations are not explained. What is u and v? where are the environmental variables taken from? Explain abbreviations. In the plots it is *S. elegans*, but in the text it's *P. elegans*.

Thank you for pointing out the need for more specific captions and clarifying abbreviations in **Figures 2 and 4**. Regarding the figure caption, we have provided more detailed descriptions of the figures' contents. We have clarified the source of the environmental variables used in the captions and corrected the species name as *P. elegans* (*Parasagitta elegans*). We have also added explanations for all abbreviations used in the figures.

Fig. 3: What is the VIP score?

The VIP (Variable Importance in Projection) score is a measure used in statistical models to estimate the importance of each variable in explaining the variation in the data. In the context of **Figure 3**, the VIP score is used to identify which ASV sequence has the most significant impact on the model's predictions. Higher VIP scores indicate ASV sequences that are more influential in the model. We included this metric to highlight key drivers in our analysis and to provide insight into which sequences are most critical for understanding the patterns observed in the data. This helps prioritize variables for further investigation or consideration in biogeochemical models. We provided more specific details related to how the VIP scores in the *Methods* section of the revised manuscript (L.437-L.440): "In the context of our analysis, higher VIP scores indicate ASV sequences that have a more substantial impact on the model's predictions, highlighting key drivers and providing insight into the most critical sequences for understanding observed patterns."

After Fig. 5, there are figures without any label and captions. The heatmaps are not readable.

We are still determining which specific figures you refer to, but we assume you mean the supplementary figures S1 and S2. In the supplementary file submitted, all figures had captions. Nevertheless, we removed labels associated with the heatmaps that were uninformative or unnecessary for clarity.

Supplement:

Which algorithms were used for denoising and chimera removal?

As mentioned in the *Methods* section (L.400-L.403), chimeras and singletons were removed from the dataset by retaining sequences in at least three samples, each with a minimum of 50 reads.

Taxonomic assignments were carried out with which program?

We believe there may be some misunderstanding, as all details regarding the taxonomic assignments in this study are provided in the subsection *Sample Treatment*, specifically under the paragraph *Illumina-Sequencing 18S rDNA & Sequence Analyses* in the supplementary file (L.50-L.70): "For further sequence processing the DADA2 R package (v. 1.18.0)¹⁶ was used. Asremoved. Taxonomic assignment of the ASVs was performed using the reference databases PR2 (v4.12.0)¹⁷ with default settings.". We would be happy to provide additional information if further clarification is needed.

L85: How many (%) of ASVs are unknown?

In the benthic and pelagic datasets, 18,340 and 6,458 ASVs were identified, respectively, of which 2,589 ASVs (14%) and 459 ASVs (7%) were not classified beyond the Kingdom level. This information was now included between parentheses in the sentence.

L90: How is this influenced?

As mentioned previously in the response to the Reviewer#2, the V4 region of the 18S rRNA gene was deliberately chosen early in the development of our time-series study to capture eukaryotic microbial biodiversity comprehensively. Previous comparisons with HPLC and light microscopy confirmed that the V4 region captures a broader range of eukaryotic taxa than other markers like the V9 region. The supplementary materials of revised manuscript now includes a section explaining this choice and discussing its implications and potential biases (L.97-L.106): "The V4 region of the 18S rDNA gene was specifically selected for this study due to its ability to provide a comprehensive representation of eukaryotic microbial biodiversity. Recent publications have highlighted the advantages of using the V4 region for biodiversity assessments, especially when compared to or combined with optical surveys for both phytoplankton and zooplankton^{4, 5}. This evidence supported the decision to focus on the V4 marker, to achieve an accurate and detailed representation of microbial communities over the extended time series. While multiple markers could provide additional insights, the need for consistency in capturing microbial diversity across time, along with the demonstrated effectiveness of the V4 region in this context, made it the most suitable choice for this long-term study."

Code availability: The URL does not work.

The URL did not work, therefore, I could not review the code.

Thank you for bringing this to our attention. As mentioned, the code is stored in a private GitHub repository, so the URL did not work. We plan to make the code publicly available once the manuscript is accepted. We appreciate your understanding and will ensure that the code is accessible for review at that time.

Reference

1. Soltwedel T, Hasemann C, Vedenin A, Bergmann M, Taylor J, Krauß F. Bioturbation rates in the deep Fram Strait: Results from in situ experiments at the arctic LTER observatory HAUSGARTEN. *J Exp Mar Biol Ecol* **511**, 1-9 (2019).
2. Anderson SR, Blanco-Bercial L, Carlson CA, Harvey EL. Role of Syndiniales parasites in depth-specific networks and carbon flux in the oligotrophic ocean. *ISME communications* **4**, ycae014 (2024).
3. Ramondenc S, *et al.* Effects of Atlantification and changing sea-ice dynamics on zooplankton community structure and carbon flux between 2000 and 2016 in the eastern Fram Strait. *Limnol Oceanogr*, (2022).
4. Weiß JF, *et al.* Unprecedented insights into extents of biological responses to physical forcing in an Arctic sub-mesoscale filament by combining high-resolution measurement approaches. *Sci Rep-Uk* **14**, 8192 (2024).
5. Weydmann-Zwolicka A, Dąbrowska AM, Mioduchowska M, Zwolicki A. Comparison of DNA metabarcoding and microscopy in analysing planktonic protists from the European Arctic. *Marine Biodiversity* **54**, 1-10 (2024).
6. Wietz M, *et al.* The polar night shift: seasonal dynamics and drivers of Arctic Ocean microbiomes revealed by autonomous sampling. *ISME Communications* **1**, 76 (2021).
7. Metfies K, *et al.* Protist communities in moored long-term sediment traps (Fram Strait, Arctic)–preservation with mercury chloride allows for PCR-based molecular genetic analyses. *Frontiers in Marine Science* **4**, 301 (2017).
8. Berry D, Widder S. Deciphering microbial interactions and detecting keystone species with co-occurrence networks. *Frontiers in microbiology* **5**, 219 (2014).
9. Soltwedel T, *et al.* Natural variability or anthropogenically-induced variation? Insights from 15 years of multidisciplinary observations at the arctic marine LTER site HAUSGARTEN. *Ecological Indicators* **65**, 89-102 (2016).
10. Soltwedel T, Grzelak K, Hasemann C. Spatial and temporal variation in deep-sea meiofauna at the LTER Observatory HAUSGARTEN in the Fram Strait (Arctic Ocean). *Diversity* **12**, 279 (2020).
11. Fryxell GA, Hasle GR. *Corethron criophilum* Castracane: its distribution and structure. *Biology of the Antarctic Seas IV* **17**, 335-346 (1971).
12. Leventer A, Dunbar RB. Diatom flux in McMurdo sound, Antarctica. *Marine Micropaleontology* **12**, 49-64 (1987).
13. Marra J, Boardman DC. Late winter chlorophyll a distributions in the Weddell Sea. *Marine ecology progress series Oldendorf* **19**, 197-205 (1984).
14. Fadeev E, *et al.* Sea ice presence is linked to higher carbon export and vertical microbial connectivity in the Eurasian Arctic Ocean. *Communications biology* **4**, 1255 (2021).
15. Hassett B, Gradinger R. Chytrids dominate arctic marine fungal communities. *Environmental microbiology* **18**, 2001-2009 (2016).
16. Callahan BJ, McMurdie PJ, Rosen MJ, Han AW, Johnson AJA, Holmes SP. DADA2: High-resolution sample inference from Illumina amplicon data. *Nature methods* **13**, 581-583 (2016).
17. Guillou L, *et al.* The Protist Ribosomal Reference database (PR2): a catalog of unicellular eukaryote small sub-unit rRNA sequences with curated taxonomy. *Nucleic acids research* **41**, D597-D604 (2012).

Reviewers comments on «Unveiling the pelagic-benthic coupling associated with the biological carbon pump in the Fram Strait (Arctic Ocean)» Reference: NCOMMS-24-22914A

This is a re-submission

Reviewer's comments in bold – Responses in black normal –

Text added to the manuscript in blue -

Reviewer(s) comments

Reviewer #1

I am mostly satisfied with the authors' revisions.

Thank you very much for your supportive feedback and for your guidance throughout the revision process. We appreciate your insights and suggestions, which were invaluable in strengthening our manuscript. Thank you again for your time and effort.

As per my original comment regarding switching back and forth between discussing Figure 3 and Figure 4 in the text, if the authors do not want to combine figures, I suggest making Figure 3B and 3C a separate figure - they can be combined into a new Figure 5. It is traditional to discuss all panels of a figure in the text before moving on to the next figure, so switching back and forth between Figure 3 and Figure 4 is confusing to readers.

We appreciate the reviewer's suggestion regarding the organization of Figures 3. We understand the potential for confusion when switching between the Figure 3 and 4 in the text. However, we feel that splitting Figure 3 into two separate figures (e.g., creating a new Figure 5) would not address the issue but would simply shift it to a later point in the text. Since the discussion of panels 3B and 3C currently occurs in the middle of the Figure 4 description, inserting a new figure here would require switching between Figures 4 and 5 before returning to Figure 4. Additionally, the analyses presented in the Figure 3 are conceptually similar, with Figure 3a focusing on pelagic data and Figure 3b on benthic data. Separating them into distinct figures would create redundancy and reduce readability.

Additionally, the github url provided for access to the code does not work: https://github.com/sramondenc/Pelagic-benthic_coupling.

Thank you for highlighting the accessibility of the GitHub URL (https://github.com/sramondenc/NCOMMS-24-22914_Pelagic_Benthic_Coupling_Arctic/). As mentioned in our previous responses, we initially intended to make the code publicly available upon acceptance of the manuscript. However, to facilitate immediate access for review and replication, we have now made the repository publicly accessible.

Reviewer #2

I think the authors have done a superb job at addressing the comments from all reviewers and the manuscript is now much improved.

Thank you very much for your encouraging words and for recognizing our efforts to address the reviewers' comments. We are grateful for your constructive feedback throughout the process, which has been invaluable in strengthening our manuscript.

I have only one comment regarding the use of V4. The limitations with the use of a single marker and particularly V4 should be addressed in more detail, particularly in light of recent work comparing the performance of multiple 18S markers (see <https://doi.org/10.1002/edn3.580>).

Thank you for your insightful comment regarding the limitations of using a single marker, specifically the V4 region of the 18S rDNA gene. We agree that the constraints of single-marker studies, including those utilizing V4, warrant further discussion. In the revised manuscript, we have now expanded on this

topic and acknowledged recent findings comparing multiple 18S markers². While we chose the V4 region for its demonstrated effectiveness in consistently capturing eukaryotic microbial biodiversity over an extended time series, we also recognize the valuable insights multiple markers could offer. With decreasing sequencing costs and the availability of refined primer-sets, it becomes feasible to integrate a multi-primer approach and more replicates in the analyses. However, when we started our study using the V4 region of the 18S was state of the art in 18S meta-barcoding. Moreover, this region is very well represented in the reference databases, which is crucial for accurate taxonomic annotation of the raw sequences. We have added this relevant information in the subsection “*Method limitations & uncertainties*” of the supplementary material, as follows (L.112 - L.116): “However, the choice of genetic markers can introduce significant biases in the biodiversity detected through eDNA metabarcoding^{1,2}. As sequencing costs decline and refined primer-sets become available, a multi-primer approach has become feasible. This strategy enhances the accuracy of the taxonomic annotation of raw sequences and helps minimize the effects of marker-specific limitations on biodiversity estimates^{1,2}.”.

Reviewer #3

Dear Dr. Simon Ramondenc and co-authors. Thank you for considering the comments I made in your revised version. In my opinion the manuscript is now more clearly written and contains a bit more background.

Thank you for your kind words. We are pleased that you find the manuscript clearer and that the additional background information enhances its quality. Your feedback has been invaluable in improving our work, and we appreciate your support throughout the revision process.

I have just a few comments left, that in my opinion still need to be addressed:

1. Negative controls: In my personal experience, even negative controls without visible bands in the gels can reveal some contamination after sequencing. Since it cannot be changed now, I strongly suggest that the authors at least add a sentence into their Material & methods part about their application of negative controls. Not reporting on this, particularly in a high impact journal, can give the false impression that negative controls are not necessary.

We appreciate your insights into the potential for contamination, even in negative controls without visible bands. In response to your suggestion, we have added a sentence to the *Materials & Methods* section to clarify our use of negative controls throughout the sequencing process. Additionally, we conduct negative controls of the sequencing data itself. Although the number of sequences detected in these negative controls is typically negligible, this step is crucial to ensure quality control. Given the low sequence counts, we did not include these controls in the raw data submission. We have updated the manuscript to include (L.389-L.392): “Negative controls are standard practice in molecular genetics, and we consistently incorporate them during extraction and PCR stages to monitor potential contamination. Agarose gels have been used to confirm the specific amplification of the 18S rDNA from our samples.”. We agree that transparency about this aspect is crucial, especially in a high-impact journal, and we hope this addition addresses your concern.

2. Replicates: In my opinion, writing that replicates are omitted based on one replicate in one study is not a good signal to the research community. The abundant taxa are usually consistent but not the rare ones. Please rephrase to make clear that replication is actually encouraged and that it would most likely strengthen your signals.

We agree that replication is a critical aspect of research that enhances the robustness of findings, especially for rare taxa. In the revised manuscript, we have rephrased the section “*Method limitations & uncertainties*” in the supplementary file to emphasize that while we presented data based on a single replicate in this study, we strongly encourage the use of replicates in future research (L.121-L.126): “Furthermore, previous studies have demonstrated that replicating PCRs and sequencing yield highly consistent results³, we acknowledge that including replicates is important for strengthening our findings,

particularly for rare taxa. In this study, replicate sequencing was not included; however, we encourage its use in future research to enhance the robustness of results. The extended timeframe of our research further ensures a clear distinction between technical and natural variability, supporting the validity of our approach.”. This approach will indeed strengthen our findings and provide a more comprehensive understanding of the community dynamics.

3. Sedimentation rates: I am still puzzled by this. If there is such a low sedimentation rate, and if we consider bioturbation, would you not expect the samples to contain significant temporal overlap? I think this still needs to be addressed clearer in the manuscript.

Low sedimentation rates combined with bioturbation could indeed lead to some temporal overlap in sediment layers, complicating direct comparisons between pelagic and benthic sequences over long timescales. However, this study aimed to minimize this overlap by focusing on surface sediments, which retain a strong signal from recent pelagic blooms, mainly if sampling was conducted shortly after a significant sedimentation event such as spring bloom.

While it is true that these surface sediments likely integrate signals over a range of years, a clear difference emerges between the sampling points, making this overlap less significant in the overall context of our study. The concern regarding sedimentation raised by Reviewer #3 highlighted the need to address this topic in the discussion of the main manuscript. Consequently, we have moved the paragraph on sedimentation rates from the *Materials and Methods* section to the *Discussion* section and made appropriate revisions. These changes improve the manuscript by more clearly conveying the limitations and rationale for our approach (L.288–L.296): “While the literature provides limited information on sedimentation rates specific to sites within the Fram Strait, it is recognized that sedimentation rates can vary locally due to depth, geographic location (e.g., eastern vs. western Fram Strait), and proximity to the marginal ice zone. However, general observations indicate that sedimentation rates in this region are relatively low (i.e., 1-2 cm per 1000 years), and there is considerable sediment mixing due to bioturbation⁴. Although we aimed to minimize the temporal overlap by focusing on surface sediments- believed to retain a strong signal from recent pelagic blooms- and by conducting benthic samples after large sedimentation events (i.e., during the summer), we cannot entirely exclude the possibility of temporal overlap.”.

4. I asked for a better introduction of the network metrics. I think terms such as ‘keystone ASV’ should be introduced at least a bit the first time they appear in the manuscript, not buried in the methods.

As recommended by Reviewer #3, we have added a brief explanation of network metrics, including 'keystone ASV,' to provide essential context before these terms appear in the Methods section. This term is first addressed in the description of the results observed in Figure 3. The revised text now reads (L.122-L.124): “From the two subnetworks that correlated positively with POC, Sn_3 and Sn_11, several ASVs were identified as keystone contributors essential for both carbon export and subnetwork stability, with stability assessed through the degree centrality of nodes (Fig. 3a).”.

5. Chimera removal is typically carried out with a dedicated algorithm (e.g. usearch / vsearch). I am not sure that your approach compensates for all chimeras.

We acknowledge the importance of using dedicated algorithms for chimera removal. In our study, we created an ASV table by combining denoised amplicon sequence variants from each sample pool, in which chimeras were predicted and removed: “An ASV table combining denoised amplicon sequence variants of each sample pool was created, and chimeras were predicted and removed.”. To further minimize the risk of false-positive associations, we also eliminated chimeras and singletons, resulting in the removal of 134 ASVs from the benthic dataset and 36 ASVs from the planktonic dataset (L.404–L.406): “To avoid the source of false-positive predicted association and singletons (i.e., 134 and 36 Amplicon Sequence Variants (ASVs) in the benthic and planktonic tables, respectively) were removed from the pelagic and benthic eDNA datasets.”. We appreciate your concern and have clarified our methodology in the revised manuscript to ensure our approach to chimera removal is more transparent

and robust. Consequently, we have revised the first referenced sentence (L.67–L.69) to read: “An ASV table combining denoised amplicon sequence variants of each sample pool was created, and chimeras were predicted by VSEARCH in de novo mode with default settings and removed from the sample files.” and removed the mention of chimeras in the second sentence, as it was not applicable.

6. Fig. 2: There is still no information on what u or v means in the caption. Please, add this.

We apologize for the oversight. We have updated the caption for Figure 2 to clarify that ‘u’ and ‘v’ represent the east-west and north-south components of wind, current, and sea-ice velocity, respectively. The sentence regarding ‘u’ and ‘v’ components in the caption read as (L.671-L.675): “All environmental variables (such as NAO for North Atlantic Oscillation, AO for Arctic Oscillation, AMO for Atlantic Multidecadal Oscillation, and ‘u’ and ‘v’ representing the east-west and north-south components of wind, current, and sea-ice velocity, respectively) included in the correlogram were described in Ramondenc et al.⁵.” We hope this revision improves the clarity of the figure.

7. The code is still not free for review. Just remember to publish it after publication. The code is still not free for review, which is why I could not review it.

Thank you for your follow-up regarding the code’s accessibility. We apologize for any confusion. We have made the repository publicly accessible to facilitate review and replication. We appreciate your understanding and are committed to ensuring that the code remains available for use and verification. Please let us know if there are any specific issues accessing the code, and we will address them promptly.

Reference

1. Clarke LJ, Beard JM, Swadling KM, Deagle BE. Effect of marker choice and thermal cycling protocol on zooplankton DNA metabarcoding studies. *Ecol Evol* 7, 873-883 (2017).
2. Zimmermann HH, Harðardóttir S, Ribeiro S. Assessing the performance of short 18S rDNA markers for environmental DNA metabarcoding of marine protists. *Environmental DNA* 6, e580 (2024).
3. Metfies K, et al. Protist communities in moored long-term sediment traps (Fram Strait, Arctic)–preservation with mercury chloride allows for PCR-based molecular genetic analyses. *Frontiers in Marine Science* 4, 301 (2017).
4. Soltwedel T, Hasemann C, Vedenin A, Bergmann M, Taylor J, Krauß F. Bioturbation rates in the deep Fram Strait: Results from in situ experiments at the arctic LTER observatory HAUSGARTEN. *J Exp Mar Biol Ecol* 511, 1-9 (2019).
5. Ramondenc S, et al. Effects of Atlantification and changing sea-ice dynamics on zooplankton community structure and carbon flux between 2000 and 2016 in the eastern Fram Strait. *Limnol Oceanogr*, (2022).